# A framework to evaluate IMEX schemes for atmospheric models

Oksana Guba[1], Mark A. Taylor[1], Andrew M. Bradley[1], Peter A. Bosler[1], and Andrew Steyer[1]

[1]Sandia Natl. Laboratories, PO Box 5800, MS 1321, Albuquerque, NM, 87175 USA

**Correspondence:** Oksana Guba (onguba@sandia.gov)

**Abstract.** We present a new evaluation framework for implicit and explicit (IMEX) Runge-Kutta timestepping schemes. The new framework uses a linearized nonhydrostatic system of normal modes. We utilize the framework to investigate stability of IMEX methods and their dispersion and dissipation for gravity, Rossby, and acoustic waves. We test the new framework on a variety of IMEX schemes and use it to develop and analyze a set of 2nd order low-storage IMEX Runge-Kutta methods with high CFL. We show that the new framework is more selective than the 2D acoustic system previously used in literature. Schemes that are stable for the 2D acoustic system are not stable for the system of normal modes.

## 1 Introduction

Differences in phase speeds between slow and fast waves in atmospheric models motivate development of timestepping schemes with an implicit component, to avoid expensive time step restrictions imposed by fast waves on explicit methods. The nonlinearity of the equations often imposes prohibitive cost on the solvers required by fully implicit methods, and hybrid implicit-explicit (IMEX) schemes that leverage the strengths of both have become common. Here, we develop a new framework for evaluating IMEX methods for atmospheric modeling.

We follow approaches from Durran and Blossey (2012); Weller et al. (2013); Lock et al. (2014); Rokhzadi et al. (2018), and others to present an evaluation framework that is simpler than a full 3D model while still containing the challenges associated with the presence of both slow and fast modes. Our framework is based on the normal mode analysis for systems introduced in Thuburn et al. (2002a, b) and Thuburn and Woollings (2005).

We focus on IMEX Runge-Kutta (RK) methods and their use in our primary application, the High Order Method Modeling Environment (HOMME) dynamical core (Dennis et al., 2012; Taylor et al., 2020). HOMME is the nonhydrostatic atmospheric dynamical core of the U. S. Dept. of Energy Exascale Earth System Model's (E3SM) (Rasch et al., 2019) atmosphere component. HOMME is formulated in Horizontally-Explicit Vertically-Implicit (HEVI) form and is well-suited for IMEX RK schemes where terms that carry vertically propagating acoustic waves are treated implicitly.

We adapt the techniques of Thuburn and Woollings (2005), hereafter TW2005, to the specific system of equations and prognostic variables in HOMME as well as other dynamical cores. Namely, we use a system of normal modes for a mass-based vertically-Lagrangian coordinate system with a Lorenz-staggered vertical discretization. We construct a spacetime operator for this system and study its properties, including stability, dispersion, and dissipation. Compared to previously used 2D acoustic system and the compressible Boussinesq equations (Durran and Blossey, 2012; Weller et al., 2013; Lock et al., 2014; Rokhzadi

et al., 2018), this system provides more complexity and more closely resembles the equations used in modern dynamical cores. It contains a full set of modes: east- and west-propagating acoustic and gravity waves and westward-propagating Rossby waves. It is linearized about a hydrostatic reference state and uses the common constant pressure boundary condition at the model top.

Using the new framework, we develop a family of second order, high CFL, low storage IMEX RK schemes and analyze their suitability for operational use in E3SM's high resolution science campaigns.

The remainder of this paper is structured as follows. In Section 2 we present the linearized system of equations associated with our formulation of the nonhydrostatic dynamics equations and compute its spacial numerical dispersion properties. Section 3 introduces the spacetime operator and describes our analysis of its numerical stability properties. In Section 4 we compare

the stability diagrams of several schemes and develop a new family of IMEX RK methods with desirable stability and storage properties. In Section 4.4 we investigate convergence of IMEX methods with respect to vertical resolution. Section 5 concludes.

## 2   Linearized system for normal modes

In this section we define the linearized system of equations that corresponds to the HOMME nonhydrostatic dynamics model. We recover analytical and numerical frequencies for spacial discretization of the system and confirm that the discretization

(which we broadly define here to include choice of prognostic variables, equation of state, boundary conditions, and staggering) is nearly optimal. Therefore, later the system can be used to investigate properties of IMEX spacetime operators.

### 2.1   Description of the system

In Thuburn et al. (2002a, b) and TW2005, the Euler equations for a dry adiabatic atmosphere are simplified to study normal modes. Various approximations about the geometry and Coriolis terms are made, and the systems are linearized about a hydro-

static reference state at rest. Furthermore, TW2005 presents such systems for different choices for thermodynamic variables, vertical coordinates, and equations of state. We use a vertical coordinate based on hydrostatic pressure (Laprise, 1992), where hybrid pressure levels are located on constant $s$ surfaces, and $s$ is the vertical Lagrangian coordinate satisfying $\dot{s} = 0$, following Lin (2004). Therefore, we adopt system (20)-(24) in TW2005 for the shallow atmosphere approximation and a Lagrangian vertical coordinate:

$$u_t = fv - \left( \frac{1}{\rho^r} \frac{\partial p}{\partial x} + \frac{\partial \phi}{\partial x} \right) \tag{1}$$

$$v_t = -fu - \left( \frac{1}{\rho^r} \frac{\partial p}{\partial y} + \frac{\partial \phi}{\partial y} \right) \tag{2}$$

$$w_t = -g \frac{\sigma}{\sigma^r} - \frac{1}{\sigma^r} \frac{\partial p}{\partial \theta} \tag{3}$$

$$\phi_t = gw \tag{4}$$

$$\sigma_t = -\sigma^r \left( \frac{\partial u}{\partial x} + \frac{\partial v}{\partial y} \right) \tag{5}$$

Here $u$, $v$, and $w$ are velocity components, $p$ is pressure, $\rho$ is density, $\phi$ is geopotential, $g$ is the gravity constant, $f$ is the Coriolis parameter, $\sigma$ is pseudo-density defined with respect to the vertical coordinate (see Taylor et al. (2020) for details), and

$\theta$ is potential temperature. The superscript $r$ denotes variables defined by reference profiles of a linearized hydrostatic steady state with constant temperature $T_0$. The subscript $t$ denotes partial differentiation with respect to time. Variables $u$, $v$, $w$, $\phi$, $p$, and $\sigma$ are first-order perturbed quantities, about the reference state, as follows from linear analysis.

After substituting single mode solutions in which each field is proportional to $\exp(ik_x x + il_y y - i\omega t)$, this formulation is equivalent to the system of equations (20)–(24) in isentropic coordinate from TW2005. With inclusion of the $\beta$-effect as in equations (55)–(56) of TW2005, this system is as follows:

$$-i\omega u = fv + \frac{ik_x}{K^2}\beta u - ik_x\left(\frac{p}{\rho^r} + \phi\right) \tag{6}$$

$$-i\omega v = -fu + \frac{ik_x}{K^2}\beta v - il_x\left(\frac{p}{\rho^r} + \phi\right) \tag{7}$$

$$-i\omega w = -g\frac{\sigma}{\sigma^r} - \frac{p_\theta}{\sigma^r} \tag{8}$$

$$-i\omega\phi = gw \tag{9}$$

$$-i\omega\sigma = -\sigma^r(ik_x u + il_x v) \tag{10}$$

with linearized equation of state (EOS)

$$\frac{p}{p^r} = \frac{1}{1-\kappa}\frac{\sigma}{\sigma^r} - \frac{1}{1-\kappa}\frac{\phi_\theta}{\phi_\theta^r}. \tag{11}$$

We also retain a version of system (6)–(10) with time derivatives in the left hand side:

$$u_t = fv + \frac{ik_x}{K^2}\beta u - ik_x\left(\frac{p}{\rho^r} + \phi\right) \tag{12}$$

$$v_t = -fu + \frac{ik_x}{K^2}\beta v - il_x\left(\frac{p}{\rho^r} + \phi\right) \tag{13}$$

$$w_t = -g\frac{\sigma}{\sigma^r} - \frac{p_\theta}{\sigma^r} \tag{14}$$

$$\phi_t = gw \tag{15}$$

$$\sigma_t = -\sigma^r(ik_x u + il_x v) \tag{16}$$

In addition to the variables and constants defined above, $\kappa = R/c_p$ is a thermodynamic constant and $k_x$ and $l_x$ are horizontal wavenumbers with $K^2 = k_x^2 + l_x^2$. Here and later in the text, the subscript $x$ in $k_x$ and $l_x$ does not denote differentiation in $x$. We keep such notations to be consistent with notations for horizontal and vertical wavenumbers introduced in Weller et al. (2013); Lock et al. (2014). The subscript $\theta$ denotes partial differentiation with respect to potential temperature. In (6)–(10) , (11), and 80  (12)–(16) variables $\rho^r$, $\sigma^r$, $p^r$, and derivative $\phi_\theta^r$ are variables defined by the reference profile of a linearized hydrostatic steady state with constant temperature $T_0$. Variables $u$, $v$, $w$, $\phi$, $p$, and $\sigma$ are first-order perturbed quantities about the reference state. All variables are scalar quantities.

We note that since $\dot{\theta} = 0$, the linear system associated with the TW2005 isentropic model is equivalent to the linear system derived with a vertically-Lagrangian coordinate. We use the same bottom boundary condition (BC) $\phi = 0$ for systems (6)–(10) 85  and (12)–(16) as in TW2005, but a different top BC. We replace the rigid lid boundary condition with a constant pressure

boundary condition, which for perturbed pressure variable becomes $p_{\text{top}} = 0$. This BC is more typical for a mass-based vertical coordinate.

We define meridional wave number $l_x = 0$, temperature of reference state $T_0 = 250$ K, depth of domain in vertical $D = 10^5$ m, Coriolis parameters $\beta = 1.619 \times 10^{-11}$ s$^{-1}$m$^{-1}$ and $f = 1.031 \times 10^{-4}$ s$^{-1}$, gravitational acceleration $g = 9.80616$ ms$^{-2}$, and thermodynamic constants $R = 287.05$ Jkg$^{-1}$K$^{-1}$ and $c_p = 1005.0$ Jkg$^{-1}$K$^{-1}$.

To study dispersion properties of system (6)–(10) we choose horizontal wavenumber $k_x = 2\pi/10^6$ m$^{-1}$ and set the number of vertical levels $n_{\text{lev}} = 20$. Dispersion and dissipation diagrams of the spacetime operators are also computed with the same $n_{\text{lev}}$ and $k_x$ to match frequencies and eigenvectors of the spacetime operators with ones from the spacial discretization.

To form a spacetime operator using Eqs. (12)–(16) and study the stability of IMEX schemes, we set $n_{\text{lev}} = 72$, to emulate the default configuration of E3SM, and vary $k_x$ throughout a representative parameter space resolvable by anticipated high resolution models. In regimes where stability is controlled by Courant-Friedrichs-Lewy (CFL) condition associated with acoustics modes, we desire an IMEX method where the stability will not depend on the number of vertical levels. In Sect. 4.4 we study the stability of IMEX schemes for varying number of vertical levels.

## 2.2   Analytic frequencies and dispersion relations

The problem of finding frequencies $\omega$ in system (6)-(10) is equivalent to investigating a spectrum of a differential operator. Since we replace boundary conditions at the top of the model, we obtain slightly different dispersion relation for internal modes compared to previous work. And, in contrast with TW2005, there are no external modes in our system.

To derive the dispersion relation from (6)–(10) , we follow Sect. 3 of TW2005 and Thuburn et al. (2002b). The dispersion relation is independent of the choice of vertical coordinate and is most easily found using the height coordinate, $z$. In TW2005 the hydrostatic equation, elimination, and use of the EOS yield the ODE, Eq. (57)

$$(\partial_z + A)(\partial_z + B)p + C = 0 \tag{17}$$

where the constants $A$ and $B$ are related to the static stability and sound speed, respectively, of the isothermal reference state and $C(\omega)$ is a cubic function of the frequency $\omega$. Expressions for $A, B$, and $C$ are defined as in TW2005, equation (58). As in TWS2002b, we make the change of variable $\tilde{p} = p \exp\left(\frac{(A+B)z}{2}\right)$ to obtain

$$\tilde{p}_{zz} + a\tilde{p} = 0, \quad a(\omega) = C(\omega) - \frac{(B-A)^2}{4}. \tag{18}$$

In our setting, ODE (18) has bottom boundary condition

$$\tilde{p}_z + \frac{B-A}{2}\tilde{p} = 0 \tag{19}$$

at $z = 0$ and top boundary condition $\tilde{p} = 0$ at $z = D$.

We first assume $a > 0$. Cases $a < 0$, $a = 0$ are discussed below. With $m = \sqrt{a}$ and solution of form $\tilde{p}(z) = c_1 \sin(mz) + c_2 \cos(mz)$, we recover internal modes. From the top boundary condition we recover

$$0 = \tilde{p}(D) = c_1 \sin(mD) + c_2 \cos(mD) \quad \Rightarrow \quad c_2 = -c_1 \tan(mD).$$

From the bottom boundary condition we recover

$$0 = \tilde{p}_z(0) + \frac{B-A}{2}\tilde{p}(0) = c_1 m + c_2 \frac{B-A}{2} \quad \Rightarrow \quad c_2 = -c_1 \frac{2m}{B-1}.$$

Combining, we obtain a condition on wavenumber $m$:

$$\tan(mD) = \frac{2m}{B-A}. \tag{20}$$

In TW2005, internal modes obey $m = n\pi/D$, where $n > 0$ is the mode number. In (20), for large $m$ wavenumbers are close to $\frac{n\pi}{D} + \frac{\pi}{2D}$, where $n$ is a positive integer.

Due to the nonlinearity of (20) with respect to $m$, wavenumbers $m$ obeying (20) are found numerically in Matlab by solving Eq. (20) for the first $n_{\text{lev}}$ values of $m_i$, $i \in \{1, \ldots, n_{\text{lev}}\}$, $n_{\text{lev}} = 20$. We recover three wave branches, acoustic, gravity, and Rossby, by solving the quintic equation

$$a(\omega) = C(\omega) - \frac{(B-A)^2}{4} = m_i^2, \tag{21}$$

that follows from substituting $a = m_i^2$ in Eq. (18), and solving for $\omega$ for each $m_i$. Three branches of internal waves are plotted as solid blue lines in Fig. 1.

External modes are derived assuming $a < 0$ in Eq. (18). Solutions are then represented by $\tilde{p}(z) = c_1 e^{mz} + c_2 e^{-mz}$, $m = \sqrt{-a}$. This leads to equation

$$\tanh(mD) = \frac{2m}{B-A} \tag{22}$$

which does not have a solution: rewrite it as $\tanh(\tilde{m}) = \frac{2\tilde{m}}{(B-A)D}$ with $\tilde{m} = mD$. Since $\frac{2}{(B-A)D} > 3$, the line $\frac{2\tilde{m}}{(B-A)D}$ and the curve $\tanh \tilde{m}$ do not intersect except at the origin. The origin is not a solution since we assumed $a < 0$ and thus $m \neq 0$. Similarly, choice $a = 0$ cannot have solutions satisfying boundary conditions for our particular value of $D$.

Analytically, one can recover external modes if depth of the domain, $D$, is bigger. We searched for values $m$ in Eq. (22) for $D = 40000$ m and $D = 50000$ m and used these values in Eq. (21). We obtained five real roots with magnitudes of order $10^{-6}$, $10^{-3}$, and $10^{-2}$, as expected. We were unable to locate external modes in discretized systems with large domain sizes. To search, we examined eigenvalues and eigenvectors (external modes have zero vertical structure in vertical velocity).

### 2.3 Numerical frequencies for HOMME discretization

To discretize the right hand side of systems (12)–(16) and (6)–(10) vertically in space, we use a Lorenz staggering and place $u$, $v$, and $\sigma$ at the midpoints of the model's $n_{\text{lev}}$ vertical levels and $\phi$ and $w$ its $n_{\text{lev}} + 1$ level interfaces. This staggering is denoted $[wz, uv\sigma]$. Due to the choice of boundary conditions, $\phi$ and $w$ are zero at the bottom of the domain; therefore, we solve only for their $n_{\text{lev}}$ interface values, excluding bottom interface. The total vector length in the discretized system is $5n_{\text{lev}}$.

Our placement of variables requires four discrete operators: one to interpolate $\phi$ from interfaces to midlevels, one to approximate the derivative $\phi_\theta$ at midlevels, one to interpolate $\sigma$ from midlevels to interfaces, and one to approximate the derivative $p_\theta$ on interfaces. Derivatives are formed using second-order finite differencing with constant level spacing, $\Delta\theta$. Interpolation

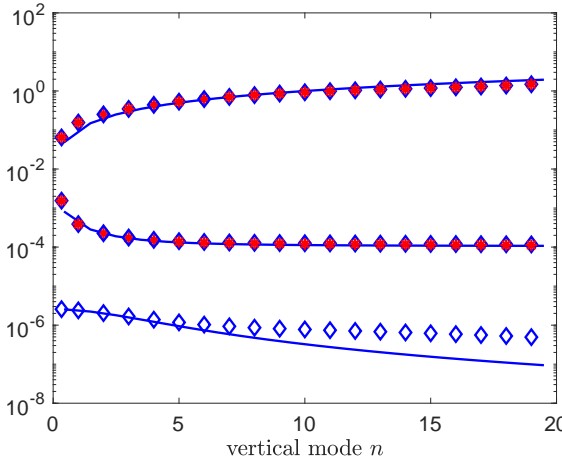

**Figure 1.** Analytical and numerical dispersion relation for system (6)–(10) . Solid blue curves, from top to bottom, are acoustic, gravity and Rossby branches of analytical solutions for $\omega$. Blue diamonds are negative numerical eigenvalues and red stars are positive numerical frequencies $\omega$.

to and from midlevels is implemented via simple averaging of neighbor values. Applying these operators at each level and interface, we can now write the discretization of system (12)–(16) as a matrix equation,

$$\boldsymbol{U}_t = \mathbf{M}\boldsymbol{U}. \tag{23}$$

Matrix $\mathbf{M}$ is of size $5n_{\text{lev}} \times 5n_{\text{lev}}$. The eigenvalues of $\mathbf{M}$ are discrete representations of quantities $-i\omega$ in (6)–(10) and eigenvectors of $\mathbf{M}$ correspond to three branches of waves, Rossby, gravity, or acoustic.

We compute the numerical eigenvalues of $\mathbf{M}$ with Matlab, then match a vertical mode to each numerical eigenvalue. To find a vertical mode, we wrote a routine to count zeros in an imaginary eigenvector part that corresponds to $w$. For the five smallest wavenumbers we diagnose $n = 1/3$ manually. A few solutions for highest wavenumbers for Rossby and gravity waves become

oscillatory and counting zeros for them is inaccurate. We diagnose them using monotonicity of numerical eigenvalues.

The numerical dispersion relation for the discretization of system (6)–(10) is plotted in Fig. 1, with blue diamonds for westward propagating waves with $\omega < 0$ and red stars for eastward propagating waves with $\omega > 0$. As in TW2005, system $[wz, uv\sigma]$, which is characterized by its staggering, choice of prognostic variables, and EOS, is in category 2b. Categories for discretizations are defined in TW2005. The most optimal category is category 1 for methods with numerical dispersion very

close to analytical. The next most optimal methods belong to the set of Category 2 methods. Category 2b methods have a near optimal dispersion relation with overestimated Rossby frequencies, as shown in Fig. 1, where numerical frequencies for the Rossby branch for large mode numbers are bigger by absolute value (Rossby frequencies are negative) than their analytical counterparts.

## 3   Stability of IMEX methods from eigenvalues of a spacetime operator

In the previous section we evaluated the properties of the spatial discretization for system (6)–(10) . We now combine the spatial discretization with a temporal discretization and then evaluate the resulting spacetime operator.

### 3.1   Spacetime operator

Similarly to Weller et al. (2013) and Lock et al. (2014), we form a spacetime operator from system (12)–(16) and compute its spectrum numerically. To be stable, the eigenvalues of the spacetime operator should lie on or inside of the unit circle.

The spacetime operator is defined by the underlying IMEX scheme. Given a linear ODE

$$\boldsymbol{y}_t = \mathbf{S}\boldsymbol{y} + \mathbf{N}\boldsymbol{y}, \tag{24}$$

where $\mathbf{S}$ and $\mathbf{N}$ are the stiff and nonstiff parts already discretized in space. The spacetime operator $\mathbf{Q}$ can be formed from the double Butcher tableau of explicit (left) and implicit (right) tables associated with a particular IMEX scheme,

$$\begin{array}{c|c} \boldsymbol{c} & \mathbf{A} \\ \hline & \boldsymbol{b}^T \end{array} \qquad \begin{array}{c|c} \hat{\boldsymbol{c}} & \hat{\mathbf{A}} \\ \hline & \hat{\boldsymbol{b}}^T \end{array}.$$

Here $\mathbf{A}$ denotes the explicit matrix for an IMEX scheme and has no relation with constant $A$ used in Sect. 2.2. We keep both notations as is due to their use in literature.

Matrices $\mathbf{A} = \{a_{j_1 j_2}\}$ and $\hat{\mathbf{A}} = \{\hat{a}_{j_1 j_2}\}$, $j_1, j_2 = 1, ..., \nu$, $\nu$ is number of stages, and vectors $\boldsymbol{c} = \{c_{j1}\}$ and $\hat{\boldsymbol{c}} = \{\hat{c}_{j1}\}$ that
determine the location of internal stages obey $c_{j_1} = \sum_{j_2} a_{j_1 j_2}$ and $\hat{c}_{j_1} = \sum_{j_2} \hat{a}_{j_1 j_2}$. Weight vectors are $\boldsymbol{b} = \{b_{j2}\}$ and $\hat{\boldsymbol{b}} = \{\hat{b}_{j2}\}$. Upper-diagonal and diagonal coefficients of the explicit matrix by definition are zero, $a_{j_1 j_2} = 0$, $j_1 \leq j_2$. We are only interested in diagonally implicit Runge-Kutta (DIRK) methods, therefore, for the implicit matrix $\hat{a}_{j_1 j_2} = 0$, $j_1 < j_2$.

Later we refer to order of accuracy conditions for IMEX schemes, as defined, for example, in Rokhzadi et al. (2018). First-order conditions are

$$\sum_{j_1} b_{j_1} = \sum_{j_1} \hat{b}_{j_1} = 1. \tag{25}$$

Second-order conditions include the 1st-order conditions and contain conditions for each table and coupling conditions for explicit and implicit tables,

$$\sum_{j_1} b_{j_1} c_{j_1} = \sum_{j_1} \hat{b}_{j_1} \hat{c}_{j_1} = \sum_{j_1} \hat{b}_{j_1} c_{j_1} = \sum_{j_1} b_{j_1} \hat{c}_{j_1} = \frac{1}{2}. \tag{26}$$

For details on how to construct a spacetime operator see Lock et al. (2014), Eq. (29) or (41), where the spacetime operator is
either a scalar or a 3x3 matrix and is called an amplification factor. Here, the spacetime operator is a $(5n_{\text{lev}}) \times (5n_{\text{lev}})$ matrix.

To form a spacetime operator from system (12)–(16) we should define its stiff and nonstiff parts. For the stiff part, represented by matrix $\mathbf{S}$, we consider the right-hand side terms of the equations for vertical velocity and geopotential,

$$-g\frac{\sigma}{\sigma^r} - \frac{p_\theta}{\sigma^r} \qquad \text{and} \qquad gw.$$

It can be shown analytically (Steyer et al., 2019) or numerically, using Matlab scripts for this project, that eigenvalues of $\mathbf{S}$ coincide with frequencies for acoustic waves. All other terms in the right-hand side of Eqs. (6)–(10) contribute to the nonstiff matrix $\mathbf{N}$. Unlike in the 2D acoustic system (Lock et al., 2014), the spectrum of $\mathbf{N}$ does not coincide with the slow modes of system (12)–(16) and $\mathbf{N}$ is not linear in $k_x$, in the sense that $\mathbf{N} \neq k_x \mathbf{N}_0$ for some constant operator $\mathbf{N}_0$.

## 3.2 Stability diagrams

To investigate the numerical stability of timestepping schemes it is common to refer to the 2D acoustics system (Weller et al., 2013; Lock et al., 2014; Steyer et al., 2019)

$$
\boldsymbol{y}_t = -ik_x \begin{pmatrix} 0 & 0 & 1 \\ 0 & 0 & 0 \\ c_s^2 & 0 & 0 \end{pmatrix} \boldsymbol{y} - ik_z \begin{pmatrix} 0 & 0 & 0 \\ 0 & 0 & 1 \\ 0 & c_s^2 & 0 \end{pmatrix} \boldsymbol{y}
\tag{27}
$$

with horizontal wave number $k_x$, vertical wave number $k_z$, $k_x = 2\pi/T_x$ and $k_z = 2\pi/T_z$ for wavelengths $T_x$ and $T_z$. In this system, the spacetime operator has three eigenvalues which we denote by $\lambda$. They are functions of $C_x$ and $C_z$, $\lambda = \lambda(C_x, C_z)$, for Courant numbers $C_x = c_s k_x \Delta t$ and $C_z = c_s k_z \Delta t$, where $c_s$ is the speed of sound and $k_x$ and $k_z$ are horizontal and vertical wavenumbers, respectively. The full stability diagram can then be plotted as a function of $C_x$ and $C_z$ or related quantities.

The relation $\lambda = \lambda(C_x, C_z)$ does not hold for system (12)–(16) and its corresponding spacetime operator. In this system, the eigenvalues are functions of three parameters, $\lambda = \lambda(k_x, \Delta t, n_{\text{lev}})$. For each $k_x$ and $\Delta t$, there are $5n_{\text{nlev}}$ eigenvalues. To study the stability properties in this three-dimensional parameter space, we consider two regimes. We first set $n_{\text{lev}} = 72$, to emulate the default configuration of E3SM, and vary $k_x$ throughout a representative parameter space resolvable by anticipated high resolution models. In the second regime, we fix $k_x$ to the highest frequency resolvable by a model with 3 km grid spacing and consider a range of vertical levels. As we are interested in IMEX methods that treat vertical acoustic waves implicitly, an ideal method should remain stable for all choices of $n_{\text{lev}}$.

In the first regime, we plot stability diagrams with horizontal wavelength $T_x$ on the horizontal axis and $\Delta t$ on the vertical axis. We vary $T_x$ from approximately 2 to 220 km and the time step range from 0.5 to 400 s. For each $k_x$ and $\Delta t$ we compose a spacetime operator that corresponds to a particular IMEX method. The operator's eigenvalues are computed numerically using one of Matlab's solvers. The largest (by magnitude) eigenvalue is saved to an array which is then plotted on a stability diagram. We declare a spacetime operator stable if its maximum eigenvalue is less than $1 + \epsilon_{\text{tol}}$, with $\epsilon_{\text{tol}} = 10^{-12}$. In our diagrams, stable regions are colored white.

## 3.3 Diagrams for dispersion and dissipation

Knowing the eigenpairs $(-i\omega_k, \boldsymbol{m}_k)$ of the space operator $\mathbf{M}$ as in Eq. (23) and the eigenpairs $(\lambda_j, \boldsymbol{q}_j)$ of the IMEX spacetime operator $\mathbf{Q}$ we recover additional properties of each IMEX scheme.

For small timesteps $\Delta t$ we expect the relationship between the space operator $\mathbf{M}$ and the spacetime operator $\mathbf{Q}$ constructed for $\Delta t$ step to be

$$\boldsymbol{q}_j \simeq \boldsymbol{m}_k \quad \text{and} \quad \lambda_j = l_j e^{-i\tilde{\omega}_j \Delta t} \tag{28}$$

for some real $l_j > 0$ and real $\tilde{\omega}_j$. We also expect each pair $(\boldsymbol{q}_j, \boldsymbol{m}_k)$ to be uniquely matched.

Ideally, $l_j = 1$ and $\tilde{\omega}_j = \omega_k$, that is, there are no dissipation or dispersion errors from timestepping. In practice, we observe at least some numerical dissipation from applying IMEX, especially for acoustic waves.

To make dissipation/dispersion diagrams, we use the Munkres algorithm (Munkres, 1957) and its Matlab implementation (Cao, 2020 (accessed March 22, 2020) to uniquely match each $\boldsymbol{q}_j$ with $\boldsymbol{m}_k$ using the cost function $-\frac{<\boldsymbol{m}_k, \boldsymbol{q}_j>}{|\boldsymbol{m}_k||\boldsymbol{q}_j|}$, where $< \cdot, \cdot >$ denotes an inner product. Then we examine corresponding eigenvalue $\lambda_j$ from the spacetime operator and compute its absolute value $l_j$ and its $\tilde{\omega}_i$ from Eq. (28). More on dissipation/dispersion diagrams is in Sect. 4.3 where we apply them for a family of IMEX methods M2.

## 4 Selectiveness of new framework

In Sect. 4.1 we provide an example of a scheme that appears to be stable for many practical choices of time steps if it is analysed with system (27) but is unstable for these timesteps if analysed with system of normal modes (12)–(16) . We also apply the new framework for two schemes presented in Giraldo et al. (2013) and Rokhzadi et al. (2018).

### 4.1 Scheme M1

In tables (29) we present a 6-stage IMEX scheme based on one of the explicit Runge-Kutta methods in Kinnmark, I. and Gray, W. (1984b). The explicit table, (29), left, is a low storage, 3nd order method with high CFL of $\sqrt{15} \approx 4$. We construct the implicit table, (29), right, using a backward Euler method for all implicit stages except the last one. The last stage is constructed to have three positive coefficients, including a nonzero coefficient on main diagonal of matrix $\hat{\mathbf{A}}$ and to obey the 2nd order convergence conditions for IMEX, (25)-(26). The method has the same time locations of explicit and implicit internal stages and is 2nd order accurate. It satisfies a stiffly accurate condition, that is, the last row of $\hat{\mathbf{A}}$ matches components of $\hat{\boldsymbol{b}}$.

$$
\begin{array}{c|cccccc|c|cccccc}
0 & 0 & 0 & 0 & 0 & 0 & 0 & 0 & 0 & 0 & 0 & 0 & 0 & 0 \\
1/5 & 1/5 & 0 & 0 & 0 & 0 & 0 & 1/5 & 0 & 1/5 & 0 & 0 & 0 & 0 \\
1/5 & 0 & 1/5 & 0 & 0 & 0 & 0 & 1/5 & 0 & 0 & 1/5 & 0 & 0 & 0 \\
1/3 & 0 & 0 & 1/3 & 0 & 0 & 0 & 1/3 & 0 & 0 & 0 & 1/3 & 0 & 0 \\
1/2 & 0 & 0 & 0 & 1/2 & 0 & 0 & 1/2 & 0 & 0 & 0 & 0 & 1/2 & 0 \\
1 & 0 & 0 & 0 & 0 & 1 & 0 & 1 & 5/18 & 5/18 & 0 & 0 & 0 & 8/18 \\
\hline
& 0 & 0 & 0 & 0 & 1 & 0 & & 5/18 & 5/18 & 0 & 0 & 0 & 8/18
\end{array}
\tag{29}
$$

### 4.1.1 Plotting details

We plot three stability diagrams for the M1 scheme: Fig. 2(a) with axes $(k_x \Delta t, k_z \Delta t)$ and Fig. 2(b) with $(T_x, \Delta t)$ axes for 2D acoustics system (27) and Fig. 2(c) with $(T_x, \Delta t)$ axes for system of normal modes (12)–(16) . In Fig. 2(a) the $x$-axis $k_x \Delta t$ varies from $10^{-4}$ to $10^{0.1}$ and the $y$-axis $k_z \Delta t$ varies from $10^{-4}$ to $10^2$. For spacing we use logarithmic scale and 100 sampling points. As in Lock et al. (2014) and Steyer et al. (2019), for each pair of $k_x \Delta t$ and $k_z \Delta t$ spacetime operator from Eq. (27) is computed using IMEX method M1 given by tables (29).

Figures 2(b) and 2(c) use the horizontal wavenumber $k_x$ that corresponds to wavelengths $T_x$. In the figures, $T_x$ varies from approximately 2 km to 220 km with logarithmic spacing for 100 sampling points. Since the acoustic system (27) requires $k_z$, for Fig. 2(b) we make $k_z$ span the range $K_0 = k_x \times [10^{-2}, 10^4]$. On the $y$-axis, the timestep varies from 0.5 s to 400 s with logarithmic spacing for 100 sampling points. For each pair of $(k_x, \Delta t)$, we compute a set of spacetime operators based on $k_z \in K_0$ via the same procedure as for Fig. 2(a). If for each $k_z$ operator is stable, then point $(T_x, \Delta t)$ is stable in Fig. 2(b). We chose to plot $T_x$ wavelengths on $x$-axes of stability diagrams instead of wavenumbers to make it easier to identify horizontal resolutions.

Figure 2(c) is generated identically to Fig. 2(b) except that its results come from the system (12)–(16) . Since its spatially discretized version is discretized in vertical direction, there is no need to define $k_z$.

Stability diagrams are not scaled by number of stages in IMEX methods.

### 4.1.2 Stability of M1

When using stability diagrams in Figs. 2(a,b) based on the 2D acoustic system, as in Lock et al. (2014), the scheme appears stable for reasonable timesteps and resolutions as indicated by the large white (stable) regions. Both figures show that stability of the IMEX scheme is the same as the stability of its explicit table, which is defined by Courant number $S_{M1} \approx 4$, as follows. In Fig. 2(a) a straight vertical line going through a point $k_x \Delta t = S_{M1}$ remains in the white region, and in Fig. 2(b) the stable (white) region lies below the straight line with slope 1, that goes through point that corresponds to values $(S_{M1}/(c_s k_x), \Delta t)$. Indeed, approximate values $T_x = 2000$ m, $\Delta t = 4$ s, $k_x = 0.0031$ m$^{-1}$, and $c_s = 317$ ms$^{-1}$, which is value of speed of sound of the constant reference state in system (12)–(16) , satisfy the last condition.

However, in Fig. 2(c), timesteps based solely on the stability of the explicit table in (29) are not stable. That is, the 2D acoustic system in Eq. (27) does not have enough complexity to indicate that the method can be unstable in practice. Compared to Eq. (27), the system of normal modes contains a full set of modes: east- and west-propagating acoustic and gravity waves and westward-propagating Rossby waves. It is linearized about a non-constant hydrostatic reference state and has commonly used constant pressure boundary condition at the model top.

### 4.2 Schemes ARK2(2,3,2)(Giraldo et al., 2013) and IMEX-SSP2(2,3,2)(Rokhzadi et al., 2018)

In Rokhzadi et al. (2018) one of ARK2(2,3,2) methods from Giraldo et al. (2013) is compared to a new scheme, IMEX-SSP2(2,3,2). The family of ARK2(2,3,2) schemes is characterized by parameter $a_{32}$ in the explicit table (Giraldo et al., 2013).

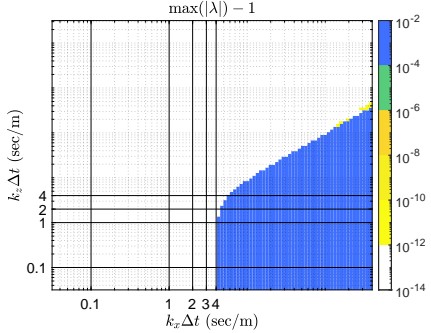

(a) 2D acoustics system, $(k_x \Delta t, k_z \Delta t)$ axes

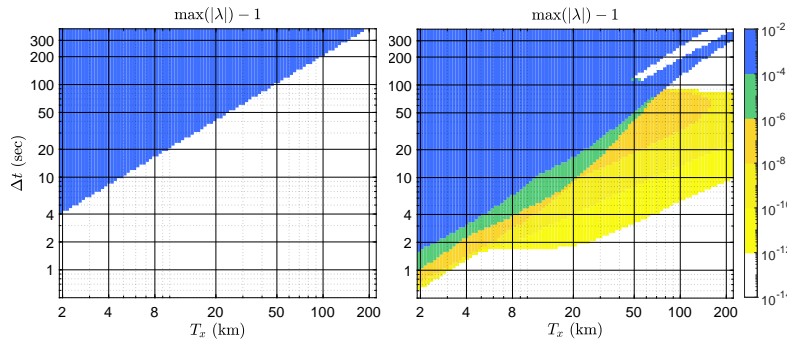

(b) 2D acoustics system, $(T_x, \Delta t)$ axes    (c) System of normal modes, $(T_x, \Delta t)$ axes

**Figure 2.** Example of selectiveness of the new framework using M1 method: stability diagram (c) based on system of normal modes deems scheme M1 as unstable for practical applications, while stable timesteps in stability diagrams (a,b) based on 2D acoustic system look acceptable.

In Rokhzadi et al. (2018) authors choose the method with $a_{32} = \frac{1}{6}(3 + 2\sqrt{2})$, which we denote here as ARK2(2,3,2)(1).
Rokhzadi et al. (2018) apply optimization to derive an ARK2 method with improved accuracy, stability, and strong stability preserving (SSP) properties as compared to ARK2(2,3,2)(1) for a linear wave equation, the 2-D acoustics system, the compressible Boussinesq equations, and the van Der Pol equation as in Durran and Blossey (2012), Weller et al. (2013), and Lock et al. (2014). We compare these two methods and method ARK2(2,3,2) with $a_{32} = 0.85$ (which we denote ARK2(2,3,2)(2)) using our system of normal modes (12)–(16) . We conclude that ARK2(2,3,2)(2) and IMEX-SSP2(2,3,2) have very similar
stability properties, as shown in Fig. 3 (b) and (c), but the stable (white) region for ARK2(2,3,2)(1) is significantly smaller, as shown in Fig. 3 (a).

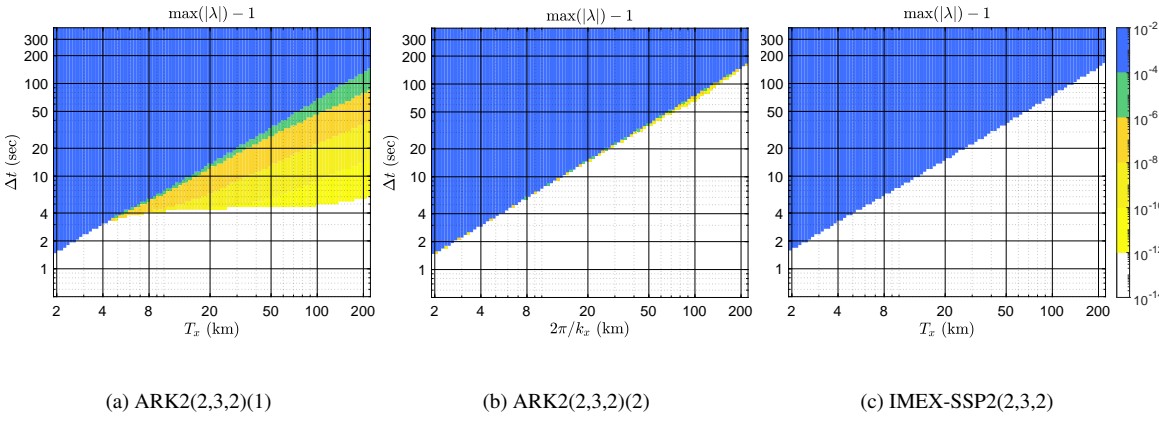

(a) ARK2(2,3,2)(1)       (b) ARK2(2,3,2)(2)       (c) IMEX-SSP2(2,3,2)

**Figure 3.** Schemes analyzed in (Rokhzadi et al., 2018) and ARK2(2,3,2)(2) scheme: (a) ARK2(2,3,2)(1) (b) ARK2(2,3,2)(2) (c) IMEX-SSP2(2,3,2)

## 4.3 Set of low storage, high CFL IMEX schemes M2

We develop a set of M2 methods using a 2nd order, explicit, low-storage, CFL of 4 Runge-Kutta scheme from Kinnmark, I. and Gray, W. (1984a). Low storage, high CFL methods developed in Kinnmark, I. and Gray, W. (1984a) and Kinnmark, I. and Gray, W. (1984b) are used in HOMME, the nonhydrostatic atmospheric dynamical core of the U. S. Dept. of Energy Exascale Earth System Model's (E3SM) atmosphere component. It is practical to extend existing explicit RK schemes to IMEX RK methods. We analyze stability, dispersion, and dissipation properties of M2 schemes using the system of normal modes (12)–(16) .

### 4.3.1 Definitions

We start with the 2nd order explicit table, (30), left, from one of methods in Kinnmark, I. and Gray, W. (1984a). For the implicit table, (30), right, we choose the same times for internal stages and make all but last implicit stages backward Euler. Internal backward Euler stages provide stability and do not affect 2nd order accuracy conditions for IMEX given by Eqs. (25)-(26).

M2 methods vary by their last implicit stage only. We require the last implicit stage to obey a stiffly accurate condition and have only nonnegative entries in its table. The last stage is defined by the vector $d$, whose entries correspond to the last row of the implicit Butcher tableau, as shown in table (30), right. Moreover, here we only consider schemes with at most three nonzero entries of $d$ due to considerations about computational cost. In practice, using IMEX method in a 3D model with

| Name | Last stage | Vector $\boldsymbol{d}$ | Order |
|------|-----------|------------------------|-------|
| M2a | | $\boldsymbol{d} = (3/11, 0, 3/11, 0, 0, 5/11)$ | 2 |
| M2b | | $\boldsymbol{d} = (0, 0, 3/5, 0, 0, 2/5)$ | 2 |
| M2c | | $\boldsymbol{d} = (2/7, 2/7, 0, 0, 0, 4/11)$ | 2 |
| M2be | backward Euler | $\boldsymbol{d} = (0, 0, 0, 0, 0, 1)$ | 1 |
| M2cn | Crank-Nicolson | $\boldsymbol{d} = (1/2, 0, 0, 0, 0, 1/2)$ | 2 |
| M2cno | Crank-Nicolson with offcentering | $\boldsymbol{d} = (1/2 - 0.02, 0, 0, 0, 0, 1/2 + 0.02)$ | 1 |

**Table 1.** Set of schemes M2

topography will require storing geopotential and vertical velocity terms for each internal stage that corresponds to $d_{j_1} \neq 0$, $j_1 < \nu$. Therefore, we focus on methods that limit such storage space.

$$
\begin{array}{c|cccccc}
0 & 0 & 0 & 0 & 0 & 0 & 0 \\
1/4 & 1/4 & 0 & 0 & 0 & 0 & 0 \\
1/6 & 0 & 1/6 & 0 & 0 & 0 & 0 \\
3/8 & 0 & 0 & 3/8 & 0 & 0 & 0 \\
1/2 & 0 & 0 & 0 & 1/2 & 0 & 0 \\
1 & 0 & 0 & 0 & 0 & 1 & 0 \\
\hline
 & 0 & 0 & 0 & 0 & 1 & 0
\end{array}
\qquad
\begin{array}{c|cccccc}
0 & 0 & 0 & 0 & 0 & 0 & 0 \\
1/4 & 0 & 1/4 & 0 & 0 & 0 & 0 \\
1/6 & 0 & 0 & 1/6 & 0 & 0 & 0 \\
3/8 & 0 & 0 & 0 & 3/8 & 0 & 0 \\
1/2 & 0 & 0 & 0 & 0 & 1/2 & 0 \\
1 & d_1 & d_2 & d_3 & d_4 & d_5 & d_6 \\
\hline
 & d_1 & d_2 & d_3 & d_4 & d_5 & d_6
\end{array}
\tag{30}
$$

where

$$
(d_1, d_2, d_3, d_4, d_5, d_6) = \boldsymbol{d}.
$$

We consider the 1st- and 2nd-order variants of M2 methods listed in Table 1.

Variants M2a, M2b, M2c (M2c is introduced in Steyer et al. (2019)) are 2nd order methods with good stability properties; their dispersive and dissipative characteristics are different, as shown below in Fig. 4. Variants M2be and M2cn are the two extremes of the M2 family. In M2be the last stage is the backward Euler method, so the scheme is expected to be the most stable but also the most dissipative method as it is 1st order accurate. Method M2cn, with Crank-Nicolson for the last stage, presumably has no dissipation for hyperbolic problems like ours. We also analyze method M2cno, where the last stage is Crank-Nicolson with off-centering, since off-centering is a common practice to stabilize timestepping schemes (Durran and Blossey, 2012; Staniforth et al., 2006).

### 4.3.2 Stability diagrams and dispersion/dissipation diagrams

Stability diagrams of the M2 schemes are shown in Figs. 4(a-c) and 5(a-c) which used plotting procedures described in Sect. 4.1.1. There, we plot numerical frequencies of space operator $\mathbf{Q}$ and spacetime operator $\mathbf{M}$ to evaluate how numerical timestepping methods, IMEX, preserve frequencies $\omega$ from space discretization $\mathbf{Q}$. In other words, due to hyperbolicity of our system

(12)–(16) exact time integration would conserve the frequencies. Inexact IMEX time integration will introduce errors, which we evaluate below. We also evaluate numerical damping introduced by IMEX since exact time integration does not introduce damping. Note that we compare properties of the spacetime operator to properties of space operator integrated exactly in time, we do not compare solutions of the spacetime operator to analytical solutions of system (12)–(16) . This is because we want to investigate the numerical errors due solely to the timestepping methods.

Dispersion/dissipation plots are shown below the stability diagrams, for the spacetime operator $\mathbf{Q}$ with eigenpairs $(\lambda_j, \boldsymbol{q}_j)$, in Figs. 4(d-f) and 5(d-f). In each figure, $\Delta t = 50\,\mathrm{s}$ and $n_{\mathrm{lev}} = 20$. There, top plots show numerical frequencies $\lambda_j$ vs vertical mode number. Red diamonds are numerical frequencies of the space operator for east- and west-propagating acoustic waves. Blue squares represent east- and west-propagating gravity waves for the space operator. Black diamonds are frequencies for west-propagating Rossby waves for the space operator. Red stars, blue plus signs, and black stars are for corresponding branches of spacetime operator. Vertical mode number and wave characterization are obtained from uniquely matching eigenvector $\boldsymbol{q}_j$ with its counterpart, eigenvector $\boldsymbol{m}_k$, of space operator $\mathbf{M}$ (also computed for 20 vertical levels).

Bottom plots in Figs. 4(d-f) and 5(d-f) show the amplification factors of eigenvalues, $|\lambda_j|$, for the spacetime operator. There, red stars, blue plus signs, and black stars are for acoustic, gravity, and Rossby waves correspondingly. Each plot shows amplification factors near 1 for gravity and Rossby waves, with additional damping of the acoustic modes. We discuss these differences further in the next section.

### 4.3.3 Analysing the M2 schemes

Due to their different final stages, the M2 schemes have different stability properties and dispersive and dissipative characteristics. To evaluate stability, we focus on regions of smallest spatial resolutions and highest wavenumbers, since those are regions where nonhydrostatic effects are most prominent. Thus stability evaluation is easy: bigger stable (white) regions translate to larger stable $\Delta t$ for those methods.

As expected, due to the presence of the last backward-Euler stage in the implicit table, the M2be scheme is the most stable. Recall that analytically for hyperbolic problems, the backward Euler method is unconditionally stable and is very dissipative. For the M2be method, the largest stable $\Delta t$ at $T_x = 2\ \mathrm{km}$ is approximately 4 sec, which is at least $2\times$ larger than the largest stable $\Delta t$ for the other schemes.

It is desirable to have an IMEX method with stability properties similar to the stability properties of an explicit method used for a non-stiff part of the system (24). In other words, it is desirable to be able to integrate a nonhydrostatic system using an IMEX method with a timestep as large as the timestep used to integrate a hydrostatic system using an explicit Runge-Kutta method. Therefore we compared the stability of IMEX method M2be with the stability of the Runge-Kutta method consisting of the explicit table in M2be (let's call this method MExplicit), when it was applied to the non-stiff part of equation (24). The stability region of M2be in Fig. 5(a) is almost as big as the stability region of MExplicit (not shown here) up to a minor difference at approximate wavelength $T_x = 220\ \mathrm{km}$. That is, the stability region of M2be is the biggest region we could possibly get from an IMEX scheme whose explicit table is part of the M2 set.

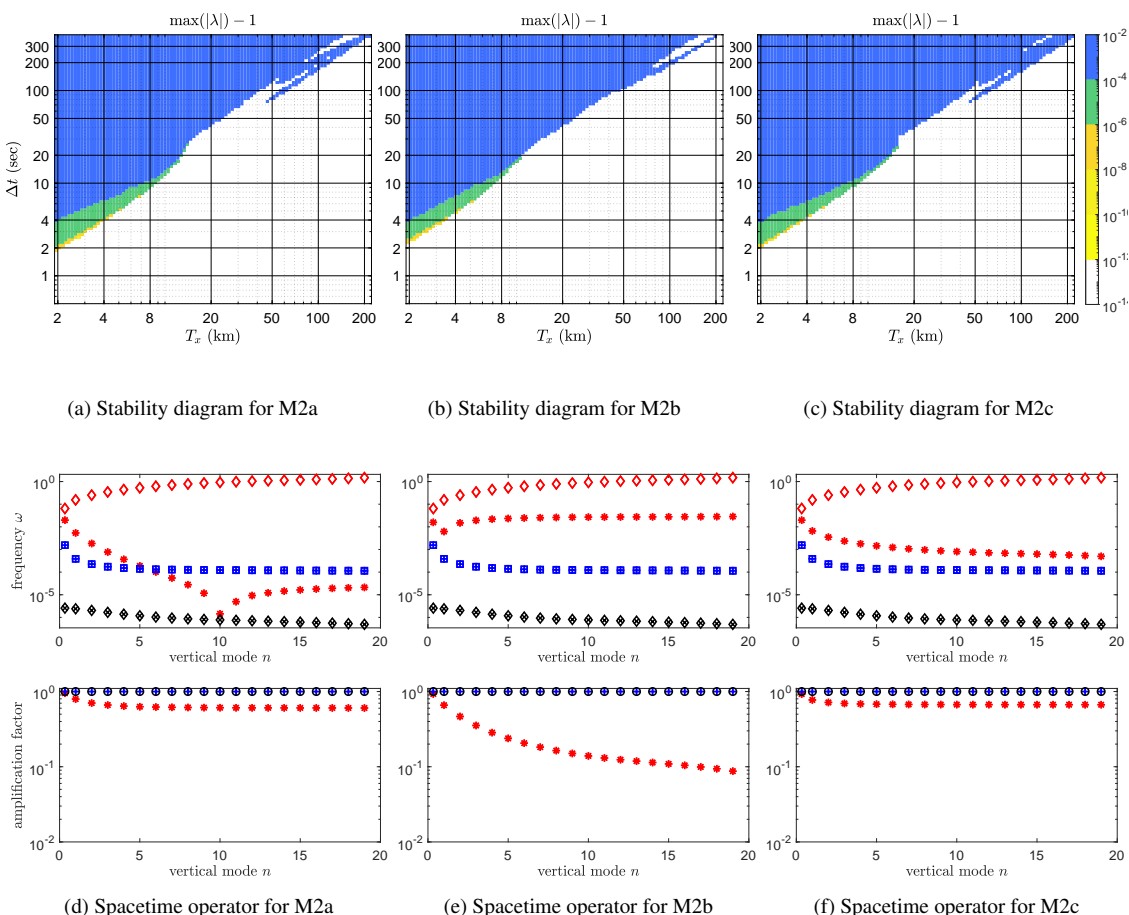

(a) Stability diagram for M2a      (b) Stability diagram for M2b      (c) Stability diagram for M2c

(d) Spacetime operator for M2a      (e) Spacetime operator for M2b      (f) Spacetime operator for M2c

**Figure 4.** Properties of M2 schemes: Stability diagrams and dispersion/dissipation

It is harder to rank schemes using dispersion/dissipation diagrams. All schemes preserve dispersion and dissipation relations for gravity and Rossby waves to a high degree. They perform very differently for acoustic waves. Method M2be has the biggest dissipation rates for acoustics waves and is the only scheme that does not have regions of negative group velocity for acoustic waves. Method M2cn is its opposite: it has no damping of acoustic waves while errors in acoustic frequencies are much larger. M2cno, a 1st order variation of M2cn, has dispersion errors very similar to M2cn while introducing low-degree dissipation into acoustic waves.

Since acoustic waves can be considered insignificant for atmospheric applications due to their low energy, one is tempted to discard numerical errors in the dispersion and dissipation of acoustic waves. However, there is an argument (Thuburn, 2012) that correct representation of even energetically weak waves in atmosphere is crucial for restoration of hydrostatic balance.

Among other 2nd order schemes, M2a, M2b and M2c, it is hard to declare a clear winner. Due to its smaller largest stable $\Delta t$ at $T_x = 2$ km and big dispersion errors, M2a may be less competitive. Comparing M2b and M2c, M2b has slightly larger

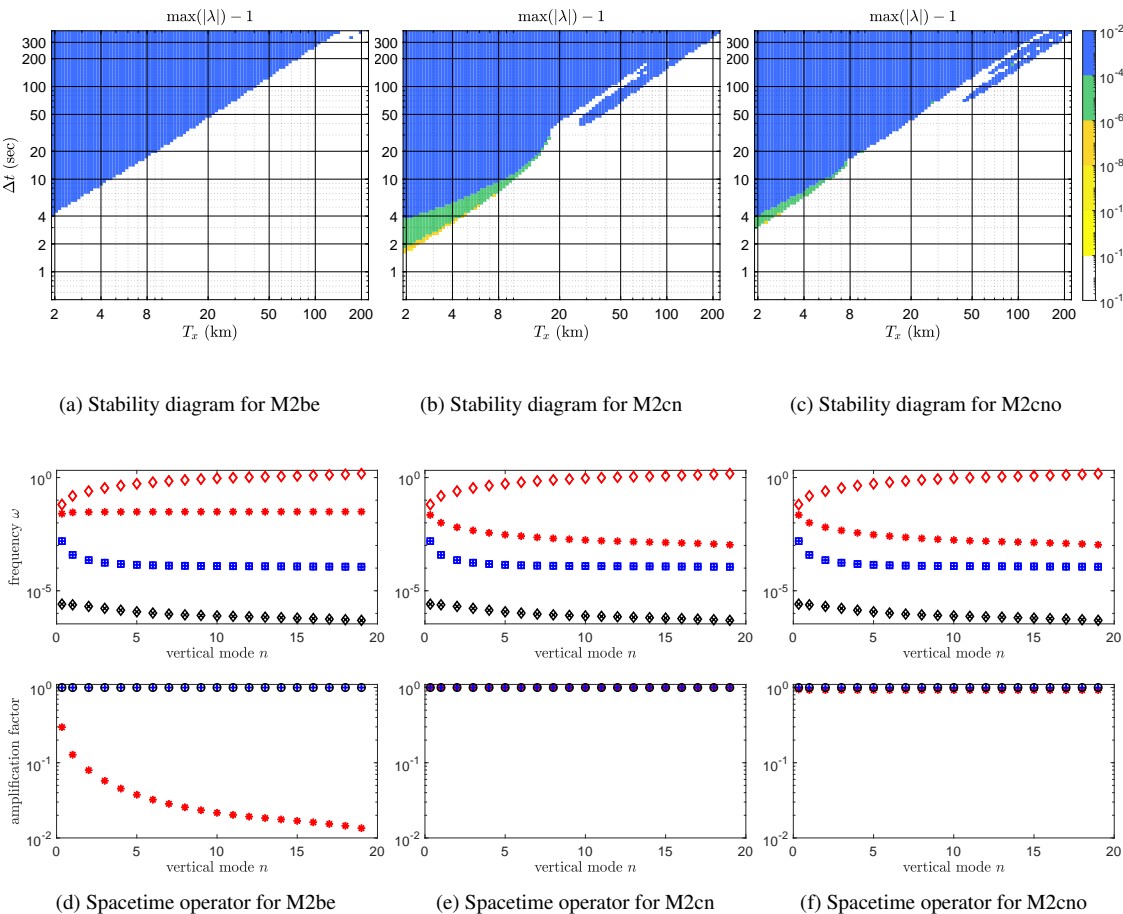

(a) Stability diagram for M2be

(b) Stability diagram for M2cn

(c) Stability diagram for M2cno

(d) Spacetime operator for M2be

(e) Spacetime operator for M2cn

(f) Spacetime operator for M2cno

**Figure 5.** Properties of M2 schemes: Stability diagrams and dispersion/dissipation

maximum stable $\Delta t$ at $T_x = 2$ km, its errors in dispersion for acoustics waves are smaller, but its dissipation rates are larger. Indeed, its dissipative rates are probably what cause its better stability compared to M2c. However, depending on evaluation criteria, M2c can be viewed as a better scheme than M2b. For example, it has smaller dissipation rates and its dispersion is very similar to Crank-Nicolson's method, which is widely used for hyperbolic problems.

### 4.3.4   Role of the implicit table

We chose to limit our search for a suitable M2 method by varying only the vector $\boldsymbol{d}$ in the implicit table. Since the 2nd order accuracy conditions for M2 family depend only on the last implicit stage, we make all other implicit stages, stages 2-5, backward Euler to presumably maximize stability. One could also try to use Crank-Nicolson or offcentered Crank-Nicolson methods for implicit stages 2-5.

To understand how the implicit stages influence dissipation of acoustic waves, we consider the expression for the final solution of Eq. (24) using Lock et al. (2014), Eq. (15), and the definition of M2 family in Eq. (30):

$$
\boldsymbol{y}^{n+1} = \boldsymbol{y}^n + \Delta t\, \boldsymbol{N}(\boldsymbol{y}^{(5)}, t^n + \Delta t) + d_1\, \Delta t\, \boldsymbol{S}\left(\boldsymbol{y}^n, t^n\right) + d_2\, \Delta t\, \boldsymbol{S}\left(\boldsymbol{y}^{(2)}, t^n + \frac{1}{4}\Delta t\right) + d_3\, \Delta t\, \boldsymbol{S}\left(\boldsymbol{y}^{(3)}, t^n + \frac{1}{6}\Delta t\right) \quad (31)
$$

$$
+ \quad d_4\, \Delta t\, \boldsymbol{S}\left(\boldsymbol{y}^{(4)}, t^n + \frac{3}{8}\Delta t\right) + d_5\, \Delta t\, \boldsymbol{S}\left(\boldsymbol{y}^{(5)}, t^n + \frac{1}{2}\Delta t\right) + d_6\, \Delta t\, \boldsymbol{S}\left(\boldsymbol{y}^{(6)}, t^n + \Delta t\right). \qquad (32)
$$

Scheme M2cn, given by its final implicit stage $(d_1, ..., d_6) = (1/2, 0, 0, 0, 0, 1/2)$, does not have dissipation. For M2cn, the solution $\boldsymbol{y}^{n+1}$ is influenced by intermediate implicit stages 2-5 only via the nonstiff term. Also, its final implicit stage is represented by the Crank-Nicolson method, known to be nondissipative for hyperbolic problems. We conclude that both of these facts contribute to the lack of dissipation in M2cn. Scheme M2be has final implicit stage $(d_1, ..., d_6) = (0, 0, 0, 0, 0, 1)$ which gives the backward Euler method. Similarly to the backward Euler method for hyperbolic problems, M2be is very dissipative.

We suggest that dispersion and dissipation of acoustics waves can be tuned by working only with the implicit table of any method.

### 4.4 Stability properties with respect to vertical resolution

For an explicit timestepping method, the most restrictive CFL condition is usually that associated with the vertically propagating acoustic waves and the stable timestep would decrease linearly with $\Delta z = D/n_{\text{lev}}$. Ideally, with an implicit treatment of vertical acoustic waves, an IMEX method should remain stable as $n_{\text{lev}}$ is increased and the stability should be controlled only by the CFL condition associated with the horizontal resolution.

To analyze this aspect of various IMEX methods, we fix $k_x$ to the highest frequency resolvable by a model with 3 km grid spacing and vary the number of vertical levels from $n_{\text{lev}} = 20$ to $n_{\text{lev}} = 100$. We plot the method's stability as a function of $n_{\text{lev}}$ using a logarithmic scale (up to rounding to the nearest integer) and 50 sampling points. Stability diagrams are made very similarly to the ones in Fig. 4, with only difference of the horizontal axis, which now defines $n_{\text{lev}}$. We vary $\Delta t$ from 1 to 10 sec with logarithmic spacing and 100 samples. Note that the horizontal axis is not defined by a vertical wavenumber, $k_z$, because for any fixed resolution $\Delta z$ the model supports waves with many vertical wavenumbers.

Figure 6 contains stability diagrams for schemes M1, ARK2(2,3,2)(1), ARK2(2,3,2)(2), and M2b. For schemes M1 and ARK2(2,3,2)(1) the stability is independent of $\Delta z$, as desired, only for up to approximately $n_{\text{lev}} = 57$ ($\Delta z \simeq 175$ m). In Figs. 6(a,b) the stable region for approximate interval $n_{\text{lev}} \in [20, 57]$ ($\Delta z \in [175, 500]$ m) is under a straight line with some $\Delta t = \Delta t_0$. For finer $\Delta z$ stability regions lie below a line with a constant slope for both schemes.

In contrast, for methods ARK2(2,3,2)(2) and M2b, stability is always controlled by the horizontal resolution: in Fig. 6(c) the stable region is below horizontal line $\Delta t_0 \simeq 7.2$ sec. To further support this conclusion, we also computed eigenvalues of the spacetime operator for method M2b, $\Delta t = 7$ sec, and a few large values of $n_{\text{lev}}$ up to 600. The spacetime operator for all large $n_{\text{lev}}$ was stable.

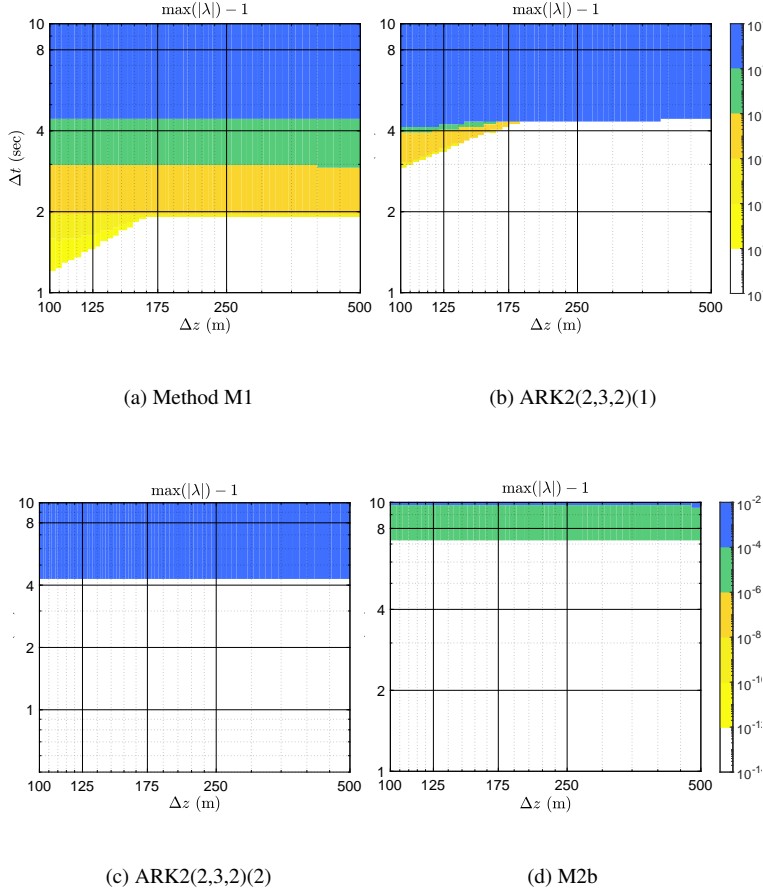

**Figure 6.** Stability diagrams with respect to varying $n_{\text{lev}}$

We do not present stability diagrams for $\Delta z$ studies for other methods from this paper because they are identical to Fig. 6(c) up to the value of $\Delta t_0$. That is, stability of methods IMEX-SSP2(2,3,2) and M2 methods is controlled by the horizontal wavelengths.

## 5 Conclusions

We developed a new framework to evaluate IMEX RK methods for atmospheric modeling. The framework uses a system of normal modes and is proven to be simple but more selective than the 2D acoustics system used in literature. For example, the M1 method from Sect. 4.1 appears to be stable for a large set of timesteps and resolutions when using the 2D acoustics system. If the method is evaluated with the system of normal modes, it is unstable for the same set of timesteps and resolutions.

The new framework gives us insight to develop a set of second-order, low storage, high CFL IMEX RK methods to use in atmospheric dynamical cores. Furthermore, we use spacetime operator built with the system of normal modes to investigate dispersion and dissipation of IMEX RK schemes for three types of waves, gravity, Rossby, and acoustic.

One extension of this work would be to investigate selectiveness of the framework based not on the system of normal modes (12)–(16) but on a system of compressible Boussinesq equations as in Durran and Blossey (2012).

*Code availability.* The current version of scripts is available from the project website: https://github.com/E3SM-Project/sta-imex under the BSD 3-clause license. The exact version of the model used to produce the results used in this paper is archived on Zenodo (Guba and Steyer, 2020 (accessed July 06, 2020). Scripts for the work presented here were written in Matlab and executed in Matlab R2018b. Description of scripts is provided in file README, which is archived with scripts at Guba and Steyer (2020 (accessed July 06, 2020). To reproduce dispersion/dissipation plots, one needs to download implementation of Munkres algorithm for Matlab separately, at Cao (2020 (accessed March 22, 2020).

For this submission, we created script `paper_figures.m`, also archived on Zenodo (Guba and Steyer, 2020 (accessed July 06, 2020), which sets parameters and launches Matlab scripts to produce all paper figures in order they appear. The script contains comments to easily identify the figures.

Sta-imex version 1.0: Copyright 2020 National Technology & Engineering Solutions of Sandia, LLC (NTESS). Under the terms of Contract DE-NA0003525 with NTESS, the U.S. Government retains certain rights in this software. For full copyright statement, see Guba and Steyer (2020 (accessed July 06, 2020).

*Author contributions.* O. Guba wrote the manuscript with contributions from P. Bosler and A. Bradley. O. Guba, M. Taylor, and A. Bradley contributed to the design of the framework. O. Guba and A. Steyer developed software for the framework.

*Competing interests.* The authors declare that they have no conflict of interest.

*Acknowledgements.* The authors thank an anonymous reviewer and Emil Constantinescu for their valuable comments and suggestions.

This research was supported as part of the Energy Exascale Earth System Model (E3SM) project, funded by the U.S. Department of Energy (DOE), Office of Science, Office of Biological and Environmental Research (BER) and by the DOE Office of Science, Advanced Scientific Computing Research (ASCR) Program under the Scientific Discovery through Advanced Computing (SciDAC 4) ASCR/BER Partnership Program.

This research was supported as part of the Energy Exascale Earth System Model (E3SM) project, funded by the U.S. Department of Energy, Office of Science, Office of Biological and Environmental Research.

Sandia National Laboratories is a multimission laboratory managed and operated by National Technology and Engineering Solutions of Sandia, LLC, a wholly owned subsidiary of Honeywell International Inc., for the U.S. Department of Energy's National Nuclear Security

Administration under Contract DE-NA0003525. This paper describes objective technical results and analysis. Any subjective views or opinions that might be expressed in the paper do not necessarily represent the views of the U.S. Department of Energy or the United States Government.

SAND2020-5447 J

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

**Appendix not to be shown**

**Deriving Eq.** (18)

Start with (Thuburn and Woollings, 2005), Eq. 57 (i did not check if (57) is correct):

$$(\partial_z + A)(\partial_z + B)p + C = 0, \text{ with BC at both ends } (\partial_z + B)p = 0,$$

where

$$A = N^2/g, \quad B = g/c^2, \quad C = \left\{ 1/c^2 + \left( K^2 + k\beta/\omega \right) \left[ f^2 - (\omega + k\beta/K^2) \right]^{-1} \right\} (\omega^2 - N^2).$$

Here $p$ seems to not be rescaled yet by any exponents. Now we try to get rid of 1st order terms: Consider $\tilde{p} = s = pe^{kz}$, $k = (A + B)/2$. Then

$$p = e^{-kz}s \tag{33}$$
$$p_z = e^{-kz}(s_z - ks) \tag{34}$$
$$p_{zz} = e^{-kz}(s_zz - 2ks_z + k^2s) \tag{35}$$

and the original eqn. becomes

$$p_{zz} + (A + B)p_z + (AB + C)p = e^{-kz}(...) = 0,$$

where what's in pars is

$$s_{zz} + s_z(-2k + (A + B)) + s(k^2 - (A + B)k + AB + C) = 0.$$

With $k = (A + B)/2$, $s_z$ multiplier is zero, $s$ multiplier is

$$-(B - A)^2/4 + C$$

and eqn. becomes

$$\tilde{p}_{zz} + \left( C - (B - A)^2/4 \right) \tilde{p} = 0$$

For BC term,

$$(\partial_z + B)p = e^{-kz}(s_z - ks + Bs) = e^{-kz}(\partial_z + (B-A)/2)s = 0,$$

that is

$$\tilde{p}_z + \frac{B-A}{2}\tilde{p} = 0.$$

## Recover int/ext modes

For now use $s = \tilde{p}$. Say, $a = m^2 > 0$, $m > 0$, $s = c_1 \sin(mz) + c_2 \cos(mz)$ and for BC then we have: For top $s = 0$ and

$$s(D) = c_1 \sin(mD) + c_2 \cos(mD) = 0$$

which leads to

$$c_2 = -c_1 \tan(mD).$$

For bottom we have

$$s_z(0) + ((B-A)/2)s(0) = c_1 m + c_2 (B-A)/2 = 0$$

and so

$$c_2 = -c_1 \frac{2m}{B-A}.$$

All together

$$-c_2 = \tan(mD) = \frac{2m}{B-A}.$$

When $m$ is large, it gets close to $pi/2 + n\pi$.

When $m$ is found numerically, we then solve

$$m^2 = a = C(\omega) - \frac{(B-A)^2}{4}$$

for $\omega$ values.

Now, case $a < 0$, take $-a = m^2$, $m > 0$: Then $s = e^m$

...

## Converting vectors to/from height coordinate

Consider function in height coordinate:

$$f(z) = f^r(z) + \epsilon \tilde{f}(z)$$

and similarly, the same function but in $\theta$ coordinate:

$$f(z) = g(\theta) = g^r(\theta) + \nu \tilde{g}(\theta).$$

Now take

$$z = z(\theta_0 + \delta\theta) = z(\theta_0) + \delta \frac{dz}{d\theta}.$$

Thus

$$f(z(\theta_0 + \delta\theta)) = f^r(z(\theta_0 + \delta\theta)) + \epsilon\tilde{f}(z)$$

and

$$f(z(\theta_0 + \delta\theta)) = f^r(z(\theta_0)) + \delta \cdot \frac{df^r}{dz} \cdot \frac{dz}{d\theta} + \epsilon\tilde{f}(z(\theta)) = g^r(\theta) + \nu\tilde{g}(\theta).$$

Comparing small terms we get

$$\frac{df^r}{dz} \cdot \frac{dz}{d\theta} + \tilde{f}(z(\theta)) = \tilde{g}(\theta).$$

Compare this to eqn 62 in TW2005 (the same). Since $w^r$ in height is zero, it is true that $w = 0$ ($w$ is perturbation) in isotermal system.

**Scalar Stability Function**

Consider IMEX with implicit and explicit time levels

$$\boldsymbol{c} = (c_1, c_2, c_3, c_4, c_5, c_6)$$

and last stage is the same as summation vector,

$$\boldsymbol{w} = (w_1, w_2, w_3, w_4, w_5, w_6).$$

In our case,

$$\boldsymbol{c} = (0, c_2, c_3, c_4, c_5, 1).$$

We are trying to derive a simplified proxy to investigate stability of IMEX. For that we assume that we integrate eqn $y_t = \lambda y$ from $y^n$ to $y^{n+1}$ with timestep $\Delta t$. Consider stages for implicit table,

$$y^{(j)}, \quad j = 1, ..., 6.$$

In our case, $y^{(1)} = y^n$ and $y^{(6)} = y^{n+1}$. Since all stages besides the 6th one are BE, we have

$$y^{(j)} = y^n + c_j \Delta t \lambda y^{(j)}$$

which is the same as

$$y^{(j)} = \frac{1}{1 - c_j \Delta t \lambda} y^n$$

for $j = 2, 3, 4, 5$. The last stage is given by

$$y^{n+1} = y^n + \Delta t \sum_{j=1}^{6} w_j \lambda y^{(j)} = y^n + \Delta t \lambda \left( w_1 y^n + w_2 \frac{1}{1 - c_2 \Delta t \lambda} + ... \right).$$

Introducing $\Delta t \lambda = s$, complex, we get

$$y^{n+1} = y^n + s \left( w_1 + \frac{w_2}{1 - c_2 s} + \frac{w_3}{1 - c_3 s} + \frac{w_4}{1 - c_4 s} + \frac{w_5}{1 - c_5 s} \right) y^n + w_6 s y^{n+1}.$$

Then

$$y^{n+1} = y^n \left(1 + sw_1 + \frac{sw_2}{1 - c_2 s} + \frac{sw_3}{1 - c_3 s} + \frac{sw_4}{1 - c_4 s} + \frac{sw_5}{1 - c_5 s}\right) / (1 - w_6 s)$$

and our stability function is

$$\left(1 + sw_1 + \frac{sw_2}{1 - c_2 s} + \frac{sw_3}{1 - c_3 s} + \frac{sw_4}{1 - c_4 s} + \frac{sw_5}{1 - c_5 s}\right) / (1 - w_6 s).$$

We can impose 2nd order conditions on $w$ and condition $\sum w_j = c_6 = 1$.

## Unwrap IMEX stages

Following ODE with 'slow' (s) and 'fast' (f) parts

$$y_t = s(y,t) + f(y,t)$$

and definition of IMEX in

`http://citeseerx.ist.psu.edu/viewdoc/download?doi=10.1.1.654.5100&rep=rep1&type=pdf`
`Runge-Kutta IMEX schemesfor the Horizontally Explicit/Vertically Implicit (HEVI) solution of`

eqns. 2-3, we use M2 explicit and implicit tables to write each stage separately:

$$
\begin{aligned}
y^{(1)} &= y^n & (36)\\
y^{(2)} &= y^n + \Delta t \frac{1}{4} s(y^{(1)}, t^n + 0) + \Delta t \frac{1}{4} f\left(y^{(2)}, t^n + \frac{1}{4}\Delta t\right) & (37)\\
y^{(3)} &= y^n + \Delta t \frac{1}{6} s\left(y^{(2)}, t^n + \frac{1}{4}\Delta t\right) + \Delta t \frac{1}{6} f\left(y^{(3)}, t^n + \frac{1}{6}\Delta t\right) ... & (38)
\end{aligned}
$$

Note that for intermediate stages like $y^{(2)}$, we solve explicit and implicit parts separately

$$
\begin{aligned}
y_1^{(2)} &= y^n + \Delta t \frac{1}{4} s(y^{(1)}, t^n + 0) & (39)\\
y^{(2)} &= y_1^{(2)} + \Delta t f\left(y^{(2)}, t^n + \frac{1}{4}\Delta t\right) & (40)
\end{aligned}
$$

The final solution is

$$
\begin{aligned}
y^{n+1} = y^{(6)} = y^n \quad &+ \quad \Delta t s\left(y^{(5)}, t^n + \Delta t\right) & (41)\\
&+ \quad d_1 \Delta t f\left(y^n, t^n\right) & (42)\\
&+ \quad d_2 \Delta t f\left(y^{(2)}, t^n + \frac{1}{4}\Delta t\right) & (43)\\
&+ \quad d_3 \Delta t f\left(y^{(3)}, t^n + \frac{1}{6}\Delta t\right) & (44)\\
&+ \quad d_4 \Delta t f\left(y^{(4)}, t^n + \frac{3}{8}\Delta t\right) & (45)\\
&+ \quad d_5 \Delta t f\left(y^{(5)}, t^n + \frac{1}{2}\Delta t\right) & (46)\\
&+ \quad d_6 \Delta t f\left(y^{(6)}, t^n + \Delta t\right) & (47)
\end{aligned}
$$