# Peer review of "A framework to evaluate IMEX schemes for atmospheric models"

_Geoscientific Model Development, 2020_

## Referee Comment (RC1) · Anonymous Referee #1 · 17 Aug 2020

**General comments:**

The authors present a new framework for the evaluation of IMEX methods for atmospheric applications where a linearized nonhydrostatic system of normal modes is used. Several IMEX methods are tested with the new framework. Tests with the new framework are compared to tests with the acoustic system. All presented IMEX methods are investigated with regard to stability. M2 methods which are a special type of IMEX methods are described. They are also investigated with regard to dispersion and dissipation properties. The authors present an intersting work on the suitability of IMEX methods for atmospheric applications which can contribute to the understanding of the numerical methods. I have the feeling that the sections where the new framework is described could contain more details. A few suggestions can be found in the section "Specific comments".

**Specific comments:**

1. section 2.1: It might help the readers if the linearized system without substituting the single mode solution was written down.

2. the same section system (6)-(7): It might contribute to a better understanding if you explained how the system (6)-(7) can be obtained from the linearized system. For better clarity, you might want to write down the whole system.

3. section 2.1 equation (8): Would you give some additional details to explain where equation (8) comes from?

4. the same section lines 91-96: Equation (9) describes a boundary condition. Could you discuss how this boundary conditions is related to $c_1, c_2$ (line 93 and 94) and equation (10)?

5. the same section line 96-98: In line 96 the wavenumbers $m$ are defined by equation (10), where $A, B$ and $D$ are constant, right? Then you write " Wavenumbers are found numerically in Matlab by solving Eq. (10) for $m_i \in \{1, \dots n_{\text{lev}}\}, n_{\text{lev}} = 20$." Why do you need the $m_i$?

6. section 2.3 line 124: The eigenvalues of $M$ are not exactly the same as the $i\omega$ in system (1)-(5).

7. section 2.3 line 132: Would you explain what category 2b is?

8. section 4.2.1 line 230-131: Can you give details about the differences between the normal mode system and the acoustic system that explain the unstable behaviour in Figure 2(c)?

9. section 4.3 line 241-242: Is there a motivation that explains why the authors use the scheme from Kinnmark, I. and Gray, W. (1984a)? Does this scheme have any specific properties?

10. section 4.3.3 line 288-289: „Its stability region in Figure 5(a) coincides with stability region of its explicit table". Would you explain which scheme is described by the explicit table? Why is the stability region of that method so large?

11. section 4.3.3 Figure 4 and Figure 5: How would you compare the methods from Figure 4 to the methods from Figure 5? Is there a difference that explains why they show up in different tables?

12. section 4.3.4 line 310: Would you explain the function $s(z)$? What does this function tell us?

13. the same section line 313-314: „we speculate that stability of any M2 method is directly related to amount of dissipation provided by the last stage coefficients $d$." Would you explain this sentence? Did you have Figure 5(a) in mind? In Figure 5(a) the stability region is large, but the dissipation rates for the acoustic waves are big. Would you say that dissipation is unaffected by the other stages?

14. section 5 Conclusion: You presented several IMEX methods. When you consider your findings, can you evaluate the different methods with respect to the applicability for atmospheric applications? How would you evaluate the M2 methods?

**Technical corrections:**

1. Figures 2-6: Please check the labelling of the color bars.

2. section 3.3 line 187: Is $\tilde{\omega}_j$ real or complex? Would you shift the tilde.

---

## Referee Comment (RC2) · Emil Constantinescu (Referee) · 23 Sep 2020

The paper is well introduced and clear in its scope and goals. I think some clarifications are in order and I have a few minor comments in reading order as follows.

1- Why are acoustic waves highlighted in the abstract and later? In particular in a HEVI implementation one hopes that the vertical acoustic waves are dissipated.

2- Formulation of the problem in Sec. 2.1: It would be great if variables are defined as scalars or vectors also please list the missing part of (6)-(7) so that this study is more self-contained and easier to follow. It should be clarified if subscripts are derivatives like in (8) or not.

3- Sec 2.2: please clarify "spectrum of ODE" on line 81. Also it may be better to spell out BC on line 108 so it's not viewed as B*C

4- Scheme ARK2(2,3,2) is misrepresented in Rokhzadi et al. (2018). The ARK2(2,3,2) is a family of methods parametrized by coefficient a_32 in the explicit part. Rokhzadi et al. (2018) picked up the one used in the numerical experiments. This was chosen for accuracy considerations: explicit part of ARK2(2,3,2) is order 3 for linear terms and it minimizes errors globally, while maintaining L-stability. The implicit part of ARK2(2,3,2) with these properties is unique (no free parameters to optimize). A closer read of Giraldo et al. (2013) reveals that a_32=0.5 provides more stability in certain regimes of the compound IMEX method. Choosing a_32=0.85 for instance produces

that corresponds to Fig. 3(a) in the manuscript and looks really similar with results in Fig 3(b), does it not?

Also,

corresponds to Fig. 6(b) in the manuscript. As a side note, ARK2(2,3,2) with a_32=0.5 has a significantly higher SSP coefficient than IMEX-SSP2(2,3,3) - explicit radius is 1.7 and implicit 2.41 as opposed to IMEX-SSP2(2,3,2). But SSP is not relevant here or in the study by Rokhzadi et al. (2018) unless a monotonic discretization is being used and discontinuous solutions develop. As Giraldo et al. (2013) note in their study they did not observe remarkable differences for ARK2(2,3,2) methods with different a_32 in practice but it may have had an impact on other regimes that were not tested exhaustively. I realize that the authors rely on the previous study. My only ask here is to acknowledge that only one method out of the ARK2(2,3,2) family has been used as it is hard to go back and fix previous published studies.

5- I cannot tell if the stability diagrams are scaled or not by the number of stages or if they should be.

---

## Author Comment (AC1) · 16 Oct 2020

We appreciate comments from the two reviewers. We addressed each comment below, listing the original comment and our reply, and in the new draft (attached). Two significant changes are expansion of Sect 2.1 and inclusion of a different ARK2(2,3,2) method, as was pointed by E. Constantinescu. We added the new method, with a32=0.85 (in explicit table), to the repository and to the draft.

5 Due to the requirement from GMDD to have "(1) comments from referees/public, (2) author's response, and (3) author's changes in manuscript" we insert old and new text in the response, not in the manuscript. This makes numbering of some formulas inconsistent with the new draft. Also, some references cannot be addressed properly, but we believe the changes are still clear.

We list Reviewers' general comments, then their individual comments in bold, then our responses and the changes we have 10 made in the manuscript. The old text is in blue, the revised text is in red.

**1 Reviewer 1**

Reviewer1 general comments:

The authors present a new framework for the evaluation of IMEX methods for atmospheric applications where a linearized nonhydrostatic system of normal modes is used. Several IMEX methods are tested with the new framework. Tests with the new framework are compared to tests with the acoustic system. All presented IMEX methods are investigated with regard to stability. M2 methods which are a special type of IMEX methods are described. They are also investigated with regard to dispersion and dissipation properties. The authors present an intersting work on the suitability of IMEX methods for atmospheric applications which can contribute to the understanding of the numerical methods. I have the feeling that the sections where the new framework is described could contain more details. A few suggestions can be found in the section "Specific comments".

**Comments 1 and 2: section 2.1: It might help the readers if the linearized system without substituting the single mode solution was written down. The same section system (6)-(7): It might contribute to a better understanding if you explained how the system (6)-(7) can be obtained from the linearized system. For better clarity, you might want to write down the whole system.**

Reply: Thank you. We added a version of system (1)-(5) before substituting the single mode solution. We also significantly extended explanation of this and other systems in Sec 2.1 to address the Reviewer's general comments. We expanded system (6)-(7) and added a clarification about it.

**OLD (blue)**

We use a vertical coordinate based on hydrostatic pressure (Laprise, 1992), where hybrid pressure levels are located on constant *s* surfaces, and *s* is the vertical Lagrangian coordinate satisfying  $\dot{s} = 0$ , following Lin (2004). After linearization and substituting single mode solutions in which each field is proportional to  $\exp(ikx + ily - i\omega t)$ , this formulation is equivalent to the system of equations (20)–(24) in isentropic coordinate from TW2005. With inclusion of the  $\beta$ -effect as in equations (55)–(56) of TW2005, this system is as follows:

$$-i\omega u = fv + \frac{ik_x}{K^2}\beta u - ik_x \left(\frac{p}{\rho^r} + \phi\right)$$
(1)

35

25

$$-i\omega v = -fu + \frac{ik_x}{K^2}\beta v - il_x \left(\frac{p}{\rho^r} + \phi\right)$$
(2)

$$-i\omega w = -g\frac{\sigma}{\sigma^r} - \frac{p_\theta}{\sigma_r}$$
(3)

$$-i\omega\phi = gw \tag{4}$$

$$-i\omega\sigma = -\sigma^r(ik_xu + il_xv) \tag{5}$$

1 . . .

~ ~ ~

$$\frac{p}{p^r} = \frac{1}{1-\kappa} \frac{\sigma}{\sigma^r} - \frac{1}{1-\kappa} \frac{\phi_\theta}{\phi_\theta^r}.$$

We also retain a version of system (14)–(18) with time derivatives in the left hand side:

$$u_t = fv + \frac{ik_x}{K^2} \beta u - ik_x \left(\frac{p}{\rho^r} + \phi\right)$$
(6)

$$\sigma_t = -\sigma^r (ik_x u + il_x v) \tag{8}$$

(7)

Here u, v, and w are velocity components, p is pressure,  $\rho$  is density,  $\phi$  is geopotential,  $k_x$  and  $l_x$  (here subscript x does not denote differentiation in x) are horizontal wavenumbers with  $K^2 = k_x^2 + l_x^2$ , g is the gravity constant,  $\kappa = R/c_p$  is a thermodynamic constant,  $\sigma$  is pseudo-density defined with respect to the vertical coordinate (see Taylor et al. (2020) for details), and  $\theta$  is potential temperature. The superscript r denotes variables defined by reference profiles of a linearized hydrostatic steady state with constant temperature  $T_0$ . The subscript t denotes partial differentiation with respect to time. The subscript  $\theta$  denotes partial differentiation with respect to potential temperature. Other variables are first-order perturbed quantities, about the reference state, as follows from linear analysis. All variables are scalar quantities.

**NEW (red)**

In Thuburn et al. (2002a, b) and TW2005, the Euler equations for a dry adiabatic atmosphere are simplified to study normal modes. Various approximations about the geometry and Coriolis terms are made, and the systems are linearized about a hydrostatic reference state at rest. Furthermore, TW2005 presents such systems for different choices for thermodynamic variables, vertical coordinates, and equations of state. We use a vertical coordinate based on hydrostatic pressure (Laprise, 1992), where hybrid pressure levels are located on constant *s* surfaces, and *s* is the vertical Lagrangian coordinate satisfying  $\dot{s} = 0$ , following Lin (2004). Therefore, we adopt system (20)-(24) in TW2005 for the shallow atmosphere approximation and a Lagrangian vertical coordinate:

65

70

40

45

50

55

$$u_t = fv - \left(\frac{1}{\rho^r}\frac{\partial p}{\partial x} + \frac{\partial \phi}{\partial x}\right) \tag{9}$$

$$v_t = -fu - \left(\frac{1}{\rho^r}\frac{\partial p}{\partial y} + \frac{\partial \phi}{\partial y}\right) \tag{10}$$

$$w_t = -g\frac{\sigma}{\sigma^r} - \frac{1}{\sigma^r}\frac{\partial p}{\partial \theta}$$
(11)

$$\phi_t = gw \tag{12}$$

$$\sigma_t = -\sigma^r \left(\frac{\partial u}{\partial t} + \frac{\partial v}{\partial t}\right) \tag{13}$$

$$\sigma_t = -\sigma^r \left( \frac{\partial x}{\partial x} + \frac{\partial y}{\partial y} \right) \tag{13}$$

Here u, v, and w are velocity components, p is pressure,  $\rho$  is density,  $\phi$  is geopotential, g is the gravity constant, f is the Coriolis parameter,  $\sigma$  is pseudo-density defined with respect to the vertical coordinate (see Taylor et al. (2020) for details), and  $\theta$  is potential temperature. The superscript r denotes variables defined by reference profiles of a linearized hydrostatic steady state with constant temperature  $T_0$ . The subscript tdenotes partial differentiation with respect to time. Variables  $u, v, w, \phi, p$ , and  $\sigma$  are first-order perturbed quantities, about the reference state, as follows from linear analysis.

After substituting single mode solutions in which each field is proportional to  $\exp(ik_x x + il_y y - i\omega t)$ , this formulation is equivalent to the system of equations (20)–(24) in isentropic coordinate from TW2005. With

inclusion of the  $\beta$ -effect as in equations (55)–(56) of TW2005, this system is as follows:

$$-i\omega u = fv + \frac{ik_x}{K^2}\beta u - ik_x \left(\frac{p}{\rho^r} + \phi\right)$$
(14)

75

80

$$-i\omega v = -fu + \frac{ik_x}{K^2}\beta v - il_x\left(\frac{p}{\rho^r} + \phi\right)$$

$$(15)$$

$$iv w = -a \frac{\sigma}{\rho_{\theta}} \frac{p_{\theta}}{\rho_r}$$

$$-i\omega w = -g \frac{1}{\sigma^r} - \frac{1}{\sigma^r}$$

$$-i\omega \phi = g w$$
(10)
(11)
(11)

$$-i\omega\sigma = -\sigma^r(ik_xu + il_xv) \tag{18}$$

with linearized equation of state (EOS)

$$\frac{p}{p^r} = \frac{1}{1-\kappa} \frac{\sigma}{\sigma^r} - \frac{1}{1-\kappa} \frac{\phi_\theta}{\phi_\theta^r} \,. \tag{19}$$

We also retain a version of system (14)–(18) with time derivatives in the left hand side:

$$u_t = fv + \frac{ik_x}{K^2}\beta u - ik_x \left(\frac{p}{\rho^r} + \phi\right)$$
(20)

$$v_t = -fu + \frac{ik_x}{K^2}\beta v - il_x \left(\frac{p}{\rho^r} + \phi\right)$$
(21)

$$w_t = -g \frac{\sigma}{\sigma^r} - \frac{p_{\theta}}{\sigma^r}$$

$$\phi_t = qw$$
(22)
(23)

90

95

$$\sigma_t = -\sigma^r(ik_x u + il_x v) \tag{24}$$

In addition to the variables and constants defined above,  $\kappa = R/c_p$  is a thermodynamic constant and  $k_x$  and  $l_x$  are horizontal wavenumbers with  $K^2 = k_x^2 + l_x^2$ . Here and later in the text, the subscript x in  $k_x$  and  $l_x$ does not denote differentiation in x. We keep such notations to be consistent with notations for horizontal and vertical wavenumbers introduced in Weller et al. (2013); Lock et al. (2014). The subscript  $\theta$  denotes partial differentiation with respect to potential temperature. In (14)–(18), (19), and (20)–(24) variables  $\rho^r$ ,  $\sigma^r$ ,  $p^r$ , and derivative  $\phi_{\mu}^{\tau}$  are variables defined by the reference profile of a linearized hydrostatic steady state with constant temperature  $T_0$ . Variables  $u, v, w, \phi, p$ , and  $\sigma$  are first-order perturbed quantities about the reference state. All variables are scalar quantities.

**- Comment 3: section 2.1 equation (8): Would you give some additional details to explain where equation (8) comes from?**

Reply: Eqn (8) is derived in references TW2005 and Thuburn et al. (2002b). We rewrote the corresponding paragraph to provide more details and the change of variable that was used.

**100 OLD (blue)**

 $\phi_t = gw$

To derive the dispersion relation, we follow Sect. 3 of TW2005 and Thuburn et al. (2002b). The dispersion relation is independent of the choice of vertical coordinate and is most easily found using the height coordinate, z. The hydrostatic equation, elimination, and use of the EOS yield the ODE

$$\tilde{p}_{zz} + a\tilde{p} = 0, \quad a(\omega) = C(\omega) - \frac{(B-A)^2}{4},$$
(25)

where  $\tilde{p} = p \exp\left(\frac{(A+B)z}{2}\right)$  is a change of variable, the constants A and B are related to the static stability and sound speed, respectively, of the isothermal reference state and  $C(\omega)$  is a cubic function of the frequency  $\omega$ . A, B, and C are defined as in TW2005, equation (58). In our setting, the ODE has bottom boundary condition

$$\tilde{p}_z + \frac{B-A}{2}\tilde{p} = 0 \tag{26}$$

at z = 0 and top boundary condition  $\tilde{p} = 0$  at z = D.

NEW (red)

To derive the dispersion relation from (14)–(18), we follow Sect. 3 of TW2005 and Thuburn et al. (2002b). The dispersion relation is independent of the choice of vertical coordinate and is most easily found using the height coordinate, *z*. In TW2005 the hydrostatic equation, elimination, and use of the EOS yield the ODE, Eq. (57)

$$(\partial_z + A)(\partial_z + B)p + C = 0 \tag{27}$$

where the constants A and B are related to the static stability and sound speed, respectively, of the isothermal reference state and  $C(\omega)$  is a cubic function of the frequency  $\omega$ . Expressions for A, B, and C are defined as in TW2005, equation (58). As in TWS2002b, we make the change of variable  $\tilde{p} = p \exp\left(\frac{(A+B)z}{2}\right)$  to obtain

120
$$\tilde{p}_{zz} + a\tilde{p} = 0, \quad a(\omega) = C(\omega) - \frac{(B-A)^2}{4}.$$
 (28)

In our setting, ODE (28) has bottom boundary condition

$$\tilde{p}_z + \frac{B-A}{2}\tilde{p} = 0 \tag{29}$$

at z = 0 and top boundary condition  $\tilde{p} = 0$  at z = D.

**- Comment 4: the same section lines 91-96: Equation (9) describes a boundary condition. Could you discuss how this boundary conditions is related to c1, c2 (line 93 and 94) and equation (10)?**

Reply: To make it more clear, we included derivation of equation (10) using coefficients  $c_1, c_2$ .

OLD (blue)

With  $m = \sqrt{a}$  and solution of form  $\tilde{p}(z) = c_1 \sin(mz) + c_2 \cos(mz)$ , we recover internal modes. Instead of internal modes with vertical wavenumber  $m = n\pi/D$ , where n > 0 is the mode number, as in TW2005, we obtain solutions with wavenumber m obeying

$$\tan(mD) = \frac{2m}{B-A} \,. \tag{30}$$

**NEW (red)**

With  $m = \sqrt{a}$  and solution of form  $\tilde{p}(z) = c_1 \sin(mz) + c_2 \cos(mz)$ , we recover internal modes. From the top boundary condition we recover

$$0 = \tilde{p}(D) = c_1 \sin(mD) + c_2 \cos(mD) \quad \Rightarrow \quad c_2 = -c_1 \tan(mD).$$

From the bottom boundary condition we recover

$$0 = \tilde{p}_z(0) + \frac{B-A}{2}\tilde{p}(0) = c_1 m + c_2 \frac{B-A}{2} \quad \Rightarrow \quad c_2 = -c_1 \frac{2m}{B-1}.$$

130

125

105

110

Combining, we obtain a condition on wavenumver m:

| 135 | $\tan(mD) = \frac{2m}{B-A} \; .$ | (31) |
|-----|----------------------------------|------|
|     | $D$ $\Pi$                        |      |

In TW2005, internal modes obey  $m = n\pi/D$ , where n > 0 is the mode number. In (31), for large m wavenumbers are close to  $\frac{n\pi}{D} + \frac{\pi}{2D}$ , where n is a positive integer.

- Comment 5: the same section line 96-98: In line 96 the wavenumbers m are defined by equation (10), where A, B and D are constant, right? Then you write " Wavenumbers are found numerically in Matlab by solving Eq. (10) for  $m_i \in 1, ..., n_{lev}, n_{lev} = 20$ ." Why do you need the  $m_i$ ?

Reply: Thank you. There was a typo in the text implying that  $m_i$  are integers. We fixed the typo and explained that m are satisfying eqn. (10) and since the equation is nonlinear due to presence of a tangent, we use a numerical method in Matlab to find first few numbers m.

OLD (blue)

145

140

160

165

Wavenumbers m are found numerically in Matlab by solving Eq. (31) for  $m_i \in \{1, ..., n_{\text{lev}}\}, n_{\text{lev}} = 20$ .

NEW (red)

Due to the nonlinearity of (31) with respect to m, wavenumbers m obeying (31) are found numerically in Matlab by solving Eq. (31) for the first  $n_{\text{lev}}$  values of  $m_i$ ,  $i \in \{1, ..., n_{\text{lev}}\}$ ,  $n_{\text{lev}} = 20$ .

- Comment 6: section 2.3 line 124: The eigenvalues of M are not exactly the same as the  $i\omega$  in system (1)-(5).
- 150 Reply: Yes, thank you. We wrote the following new (red) wording about eigenvalues of matrix M.

OLD (blue)

To discretize (20)–(24) vertically in space, we use a Lorenz staggering and place...

The eigenvalues of M are  $-i\omega$  and its eigenvectors correspond to three branches of waves, Rossby, gravity or acoustic.

155 NEW (red)

To discretize the right hand side of systems (20)–(24) and (14)–(18) vertically in space, we use a Lorenz staggering and place u, v, and  $\sigma$  at the midpoints of the model's  $n_{\text{lev}}$  vertical levels and  $\phi$  and w its  $n_{\text{lev}} + 1$  level interfaces.

The eigenvalues of M are discrete representations of quantities  $-i\omega$  in (14)–(18) and eigenvectors of M correspond to three branches of waves, Rossby, gravity, or acoustic.

**- Comment 7: section 2.3 line 132: Would you explain what category 2b is?**

Reply: We clarified how Category 2b is defined and explained why our method is Category 2b.

OLD (blue)

The numerical dispersion relation for the discretization of system (14)–(18) is plotted on Fig. ?? with blue diamonds for westward propagating waves with  $\omega < 0$  and red stars for eastward propagating waves with  $\omega > 0$ . As in TW2005, system  $[wz, uv\sigma]$  which is characterized by its staggering, choice of prognostic variables and EOS, is in category 2b. This category has a near optimal dispersion relation with overestimated Rossby frequencies.

NEW (red)

170 The numerical dispersion relation for the discretization of system (14)–(18) is plotted in Fig. ??, with blue diamonds for westward propagating waves with  $\omega < 0$  and red stars for eastward propagating waves with  $\omega > 0$ . As in TW2005, system  $[wz, uv\sigma]$ , which is characterized by its staggering, choice of prognostic variables, and EOS, is in category 2b. Categories for discretizations are defined in TW2005. Categories for discretizations are defined in TW2005. The most optimal category is category 1 for methods with numerical dispersion very close to analytical. The next most optimal methods belong to the set of Category 2 methods. Category 2b methods have a near optimal dispersion relation with overestimated Rossby frequencies, as shown in Fig. **??**, where numerical frequencies for the Rossby branch for large mode numbers are bigger by absolute value (Rossby frequencies are negative) than their analytical counterparts.

180

185

190

195

175

**- Comment 8: section 4.2.1 line 230-131: Can you give details about the differences between the normal mode system and the acoustic system that explain the unstable behaviour in Figure 2(c)?**

Reply: We added text to highlight differences between the 2d acoustic system and the system of normal modes. The system of normal modes has all waves presented in the atmosphere, Rossby, gravity and acoustic, and is linearized around non-constant hydrostatic profile. It has realistic boundary conditions at the top and the bottom of the domain.

**OLD (blue)**

That is, Eq. (??) does not have enough complexity to indicate that the method can be unstable in practice. NEW (red)

That is, the 2D acoustic system in Eq. (??) does not have enough complexity to indicate that the method can be unstable in practice. Compared to Eq. (??), the system of normal modes contains a full set of modes: east- and west-propagating acoustic and gravity waves and westward-propagating Rossby waves. It is linearized about a non-constant hydrostatic reference state and has commonly used constant pressure boundary condition at the model top.

- Comment 9 (also addressing Comment 14): section 4.3 line 241-242: Is there a motivation that explains why the authors use the scheme from Kinnmark, I. and Gray, W. (1984a)? Does this scheme have any specific properties?

Reply: Kinnmark and Gray's methods are attractive due to their low storage and high CFL. We added text about this in Sec. 4.3

OLD (blue)

We develop a set of M2 methods using a 2nd order, explicit, low-storage, CFL of 4 Runge-Kutta scheme from Kinnmark, I. and Gray, W. (1984a). We analyze M2 schemes using system of normal modes (20)–(24). NEW (red)

- 200 We develop a set of M2 methods using a 2nd order, explicit, low-storage, CFL of 4 Runge-Kutta scheme from Kinnmark, I. and Gray, W. (1984a). Low storage, high CFL methods developed in Kinnmark, I. and Gray, W. (1984a) and Kinnmark, I. and Gray, W. (1984b) are used in HOMME, the nonhydrostatic atmospheric dynamical core of the U. S. Dept. of Energy Exascale Earth System Model's (E3SM) atmosphere component. It is practical to extend existing explicit RK schemes to IMEX RK methods. We analyze stability, dispersion, and dissipation properties of M2 schemes using the system of normal modes (20)–(24).
  - Comment 10: section 4.3.3 line 288-289: "Its stability region in Figure 5(a) coincides with stability region of its explicit table". Would you explain which scheme is described by the explicit table? Why is the stability region of that method so large?

Reply: Thank you. We added an explanation about 'explicit table' methods and a comment on how the last backward Euler stage (an unconditionally stable timestepping method for hyperbolic problems) makes method M2be most stable among M2 methods.

**OLD (blue)**

As expected, M2be scheme is the most stable. Its largest stable  $\Delta t$  at  $T_x = 2$  km is approximately 4 sec, which is at least  $2 \times$  larger than the largest stable  $\Delta t$  for the other schemes. Its stability region in Fig. ??(a) coincides with stability region of its explicit table (not shown here) up to a minor difference at approximate

215

wavelength  $T_x = 220$  km. That is, the stability region of M2be is the biggest region we could possibly get from an IMEX scheme whose explicit table is part of the M2 set.

NEW (red)

220

225

230

235

240

250

255

As expected, due to the presence of the last backward-Euler stage in the implicit table, the M2be scheme is the most stable. Recall that analytically for hyperbolic problems, the backward Euler method is unconditionally stable and is very dissipative. For the M2be method, the largest stable  $\Delta t$  at  $T_x = 2$  km is approximately 4 sec, which is at least  $2 \times$  larger than the largest stable  $\Delta t$  for the other schemes.

It is desirable to have an IMEX method with stability properties similar to the stability properties of an explicit method used for a non-stiff part of the system (??). In other words, it is desirable to be able to integrate a nonhydrostatic system using an IMEX method with a timestep as large as the timestep used to integrate a hydrostatic system using an explicit Runge-Kutta method. Therefore we compared the stability of IMEX method M2be with the stability of the Runge-Kutta method consisting of the explicit table in M2be (let's call this method MExplicit), when it was applied to the non-stiff part of equation (??). The stability region of M2be in Fig. ??(a) is almost as big as the stability region of MExplicit (not shown here) up to a minor difference at approximate wavelength  $T_x = 220$  km. That is, the stability region of M2be is the biggest region we could possibly get from an IMEX scheme whose explicit table is part of the M2 set.

- Comment 11: section 4.3.3 Figure 4 and Figure 5: How would you compare the methods from Figure 4 to the methods from Figure 5? Is there a difference that explains why they show up in different tables?

Reply: Could we assume you meant 'show up in different figures', not 'in different tables'? Both Figure 4 and 5 contain only methods M2. Methods M2 are defined in table 1. We used two figures for methods M2 because of space that plots need to occupy; it is not possible to fit all plots for M2 methods into one figure.

- Comment 12 and 13: section 4.3.4 line 310: Would you explain the function s(z)? What does this function tell us? the same section line 313-314: "we speculate that stability of any M2 method is directly related to amount of dissipation provided by the last stage coefficients d." Would you explain this sentence? Did you have Figure 5(a) in mind? In Figure 5(a) the stability region is large, but the dissipation rates for the acoustic waves are big. Would you say that dissipation is unaffected by the other stages?

Reply: Thank you. We clarified conclusions in affected paragraphs. We realized that discussion about s(z) is not adding to discussion and removed the notation.

Re: "We speculate that stability of any M2 method is directly related to amount of dissipation provided by the last stage coefficients d." – while this is true for the family of M2 methods, where we only vary the last implicit stage, this statement does not provide any insight; and we removed it.

Re: "Would you say that dissipation is unaffected by the other stages?" – in the revised text we gave some clarification to this question using schemes M2be and M2cn as examples. Thank you again for these questions.

OLD (blue)

We choose to limit our search for a good M2 method by varying only the vector d in the implicit table. If one wants to perform a more comprehensive search for additional members of the M2 family of schemes with explicit table from (??), left, the first step would be to focus on stability and dispersive and dissipative properties of the implicit table (??), right. In this case it is standard to form a function s(z), where z is complex and often chosen to be purely imaginary due to strong hyperbolicity of systems of atmospheric dynamics. Here, since 2nd order accuracy conditions for methods M2 depend only on the last implicit stage, we make all other implicit stages backward Euler to presumably maximize stability.

As a next step, from observing the stability and dispersion/dissipation diagrams in Figs. ??, ?? we speculate that stability of any M2 method is directly related to amount of dissipation provided by the last stage coefficients d. That is, choices where |s(z)| is smaller would lead to bigger stability regions in stability diagrams.

In the M2 methods, dispersion and dissipation of gravity and Rossby waves do not seem to be affected by the implicit table, in particular, they seem insensitive to the most dissipative backward Euler stages 2 to 5. On the contrary, acoustic waves are affected by the implicit table and its last, 6th, implicit stage. We make a suggestion that dispersion and dissipation of acoustics waves can be tuned only by working with the implicit table of any method.

265 NEW (red)

260

285

295

300

We chose to limit our search for a suitable M2 method by varying only the vector d in the implicit table. Since the 2nd order accuracy conditions for M2 family depend only on the last implicit stage, we make all other implicit stages, stages 2-5, backward Euler to presumably maximize stability. One could also try to use Crank-Nicolson or offcentered Crank-Nicolson methods for implicit stages 2-5.

To understand how the implicit stages influence dissipation of acoustic waves, we consider the expression for the final solution of Eq. (??) using Lock et al. (2014), Eq. (15), and the definition of M2 family in Eq. (??):

$$y^{n+1} = y^{n} + \Delta t \, \mathbf{N}(y^{(5)}, t^{n} + \Delta t) + d_{1} \, \Delta t \, \mathbf{S}(y^{n}, t^{n}) + d_{2} \, \Delta t \, \mathbf{S}\left(y^{(2)}, t^{n} + \frac{1}{4}\Delta t\right) + d_{3} \, \Delta t \, \mathbf{S}\left(y^{(3)}, t^{n} + \frac{1}{6}\Delta t^{2}\right) + d_{4} \, \Delta t \, \mathbf{S}\left(y^{(4)}, t^{n} + \frac{3}{8}\Delta t\right) + d_{5} \, \Delta t \, \mathbf{S}\left(y^{(5)}, t^{n} + \frac{1}{2}\Delta t\right) + d_{6} \, \Delta t \, \mathbf{S}\left(y^{(6)}, t^{n} + \Delta t\right) \,.$$
(33)

- Scheme M2cn, given by its final implicit stage (d1,...,d6) = (1/2,0,0,0,0,1/2), does not have dissipation. For M2cn, the solution yn+1 is influenced by intermediate implicit stages 2-5 only via the nonstiff term. Also, its final implicit stage is represented by the Crank-Nicolson method, known to be nondissipative for hyperbolic problems. We conclude that both of these facts contribute to the lack of dissipation in M2cn. Scheme M2be has final implicit stage (d1,...,d6) = (0,0,0,0,0,1) which gives the backward Euler method.
  Similarly to the backward Euler method for hyperbolic problems, M2be is very dissipative. We suggest that dispersion and dissipation of acoustics waves can be tuned by working only with the implicit table of any method.
  - Comment 14: section 5 Conclusion: You presented several IMEX methods. When you consider your findings, can you evaluate the different methods with respect to the applicability for atmospheric applications? How would you evaluate the M2 methods?

Reply: This is also addressed in Comment 9. Part of our work focused on family of M2 methods because its explicit table is one of RK methods used in HOMME. It is reasonable to extend the current explicit scheme in the dycore (in our case, HOMME) to an IMEX method, to keep desirable properties of the explicit method, in particular, low storage and high CFL, and to maintain common software framework for hydrostatic and nonhydrostatic models.

**- Technical corrections from Reviewer 1, Comment 1: Figures 2-6: Please check the labelling of the color bars.**

Reply: We are not sure what's wrong with color bars; they have value labels.

**- Technical corrections from Reviewer 1, Comment 2: section 3.3 line 187: Is $\tilde{\omega}_j$ real or complex? Would you shift the tilde.**

Reply: Thanks, fixed.

**OLD (blue)**

For small timesteps  $\Delta t$  we expect the relationship between the space operator **M** and the spacetime operator **Q** constructed for  $\Delta t$  step to be

$$q_j \simeq m_k$$
 and  $\lambda_j = l_j e^{-i\tilde{\omega}_j \Delta t}$  (34)

for some real  $l_j > 0$  and complex  $\tilde{\omega_j}$ . NEW (red)

... for some real  $l_j > 0$  and real  $\tilde{\omega}_j$ .

**2 Reviewer 2**

310

315

325

Reviewer 2 general comments:

The paper is well introduced and clear in its scope and goals. I think some clarifications are in order and I have a few minor comments in reading order as follows.

**- Comment 1: Why are acoustic waves highlighted in the abstract and later? In particular in a HEVI implementation one hopes that the vertical acoustic waves are dissipated.**

Reply: We tend to disagree that the best scenario is when acoustic waves are heavily dissipated. As noted by Thuburn, proper representation of all waves, including energetically insignificant acoustic and gravity waves, contributes to restoration of the hydrostatic balance in the atmosphere. This is why in M2 family we presented more dissipative (M2be) and less dissipative methods. However, we are still in search for setups or idealized tests where strong or weak dissipation of acoustic waves, or their dispersive properties, would make a difference.

**- Comment 2: Formulation of the problem in Sec. 2.1: It would be great if variables are defined as scalars or vectors also please list the missing part of (6)-(7) so that this study is more self-contained and easier to follow. It should be clarified if subscripts are derivatives like in (8) or not.**

Reply: Thank you. We added more details about subscripts and scalars (all variables presented in the systems in Sec 2.1 are scalars); this overlaps with the reply and changes for Reviewer 1, Comment 1.

- Comment 3: Sec 2.2: please clarify "spectrum of ODE" on line 81. Also it may be better to spell out BC on line 108 so it's not viewed as B\*C
- Reply: Thanks. We replaced ODE with 'differential operator'. We replaced BC on line 81 with 'boundary conditions'.

**OLD (blue)**

The problem of finding frequencies  $\omega$  in system (14)-(18) is equivalent to investigating a spectrum of an ODE. NEW (red)

The problem of finding frequencies  $\omega$  in system (14)-(18) is equivalent to investigating a spectrum of a differential operator.

**OLD (blue)**

Similarly, choice a = 0 cannot have solutions satisfying BC for our particular value of D. NEW (red) Similarly, choice a = 0 cannot have solutions satisfying boundary conditions for our particular value of D.

330 - Comment 4: Scheme ARK2(2,3,2) is misrepresented in Rokhzadi et al. (2018). The ARK2(2,3,2) is a family of methods parametrized by coefficient a32 in the explicit part. Rokhzadi et al. (2018) picked up the one used in the numerical experiments. This was chosen for accuracy considerations: explicit part of ARK2(2,3,2) is order 3 for linear terms and it minimizes errors globally, while maintaining L-stability. The implicit part of ARK2(2,3,2) with these properties is unique (no free parameters to optimize). A closer read of Giraldo et al. (2013) reveals that a32=0.5 provides more stability in certain regimes of the compound IMEX method. Choosing a32=0.85 for 335 instance produces that corresponds to Fig. 3(a) in the manuscript and looks really similar with results in Fig 3(b), does it not? Also, corresponds to Fig. 6(b) in the manuscript. As a side note, ARK2(2,3,2) with a32=0.5 has a significantly higher SSP coefficient than IMEX-SSP2(2,3,3) - explicit radius is 1.7 and implicit 2.41 as opposed to IMEX-SSP2(2,3,2). But SSP is not relevant here or in the study by Rokhzadi et al. (2018) unless a monotonic 340 discretization is being used and discontinuous solutions develop. As Giraldo et al. (2013) note in their study they did not observe remarkable differences for ARK2(2,3,2) methods with different a32 in practice but it may have had an impact on other regimes that were not tested exhaustively. I realize that the authors rely on the previous

Figure 1. Schemes analyzed in (Rokhzadi et al., 2018) and ARK2(2,3,2)(2) scheme: (a) ARK2(2,3,2)(1) (b) ARK2(2,3,2)(2) (c) IMEX-SSP2(2,3,2)

**study. My only ask here is to acknowledge that only one method out of the ARK2(2,3,2) family has been used as it is hard to go back and fix previous published studies.**

Reply: Thank you. We added method ARK2(2,3,2) with a32=0.85 to the manuscript. As you pointed out, stability plots look good for this method. We modified Figure 3 and Figure 6 with plots as shown below. Figure 3 now has new plot (b). Figure 6 also has new plot (c). We also found a typo in line 337 (original draft) and fixed it, 'M1' should be 'M2b'.

We also modified corresponding text and the repository, and updated Zenodo record for it. Thanks again for pointing this out.

| 350 | OLD (blue) |
|-----|------------|
|-----|------------|

345

355

In Rokhzadi et al. (2018) the ARK2(2,3,2) method from Giraldo et al. (2013) is compared to a new scheme, IMEX-SSP2(2,3,2). Rokhzadi et al. (2018) apply optimization to derive an ARK2 method with improved accuracy, stability, and strong stability preserving (SSP) properties as compared to ARK2(2,3,2) for a linear wave equation, the 2-D acoustics system, the compressible Boussinesq equations, and the van Der Pol equation as in Durran and Blossey (2012), Weller et al. (2013), and Lock et al. (2014). We compare the two methods using our system of normal modes (20)–(24) and similarly conclude that IMEX-SSP2(2,3,2) is more stable: the stable (white) region in stability diagram for ARK2(2,3,2) in Fig. 1 (a) is significantly smaller that the stable region for IMEX-SSP2(2,3,2) in Fig. 1 (b).

- NEW (red)
- 360In Rokhzadi et al. (2018) one of ARK2(2,3,2) methods from Giraldo et al. (2013) is compared to a new
scheme, IMEX-SSP2(2,3,2). The family of ARK2(2,3,2) schemes is characterized by parameter  $a_{32}$  in the
explicit table (Giraldo et al., 2013). In Rokhzadi et al. (2018) authors choose the method with  $a_{32} = \frac{1}{6}(3 + 2\sqrt{2})$ , which we denote here as ARK2(2,3,2)(1). Rokhzadi et al. (2018) apply optimization to derive an
ARK2 method with improved accuracy, stability, and strong stability preserving (SSP) properties as com-
pared to ARK2(2,3,2)(1) for a linear wave equation, the 2-D acoustics system, the compressible Boussinesq
equations, and the van Der Pol equation as in Durran and Blossey (2012), Weller et al. (2013), and Lock
et al. (2014). We compare these two methods and method ARK2(2,3,2) with  $a_{32} = 0.85$  (which we denote
ARK2(2,3,2)(2)) using our system of normal modes (20)–(24). We conclude that ARK2(2,3,2)(2) and IMEX-
SSP2(2,3,2) have very similar stability properties, as shown in Fig. 1 (b) and (c), but the stable (white) region
for ARK2(2,3,2)(1) is significantly smaller, as shown in Fig. 1 (a).

---

## Author Comment (AC2) · 30 Oct 2020

One of your comments was about plots' labels. After submitting a response to reviews I noticed that the labels are cropped improperly. To avoid confusion, I am not attaching the new draft with fixed figures unless requested. Thank you!
* * *